# When Should Reinforcement Learning Use Causal Reasoning?

**Oliver Schulte** *oschulte@cs.sfu.ca*

*School of Computing Science, Simon Fraser University, Vancouver, Canada*

**Pascal Poupart** *ppoupart@uwaterloo.ca*
*Cheriton School of Computer Science, University of Waterloo, Waterloo, Canada*
*Vector Institute, Toronto, Canada*

**Reviewed on OpenReview:** *https://openreview.net/forum?id=D1PPuk8ZBI*

## Abstract

Reinforcement learning (RL) and causal reasoning naturally complement each other. The goal of causal reasoning is to predict the effects of interventions in an environment, while the goal of reinforcement learning is to select interventions that maximize the rewards the agent receives from the environment. Reinforcement learning includes the two most powerful sources of information for estimating causal relationships: temporal ordering and the ability to act on an environment. This paper provides a theoretical study examining which reinforcement learning settings we can expect to benefit from causal reasoning, and how. According to our analysis, the key factor is *whether the behavioral policy—which generates the data—can be executed by the learning agent*, meaning that the observation signal available to the learning agent comprises all observations used by the behavioral policy. Common RL settings with behavioral policies that are executable by the learning agent include on-policy learning and online exploration, where the learning agent uses a behavioral policy to explore the environment. Common RL settings with behavioral policies that are not executable by the learning agent include offline learning with a partially observable state space and asymmetric imitation learning where the demonstrator has access to more observations than the imitator. Using the theory of causal graphs, we show formally that when the behavioral policy is executable by the learning agent, conditional probabilities are causal, and can therefore be used to estimate expected rewards as done in traditional RL. However, when the behavioral policy is not executable by the learning agent, conditional probabilities may be confounded and provide misleading estimates of expected rewards. For confounded settings, we describe previous and new methods for leveraging causal reasoning.

## 1 Introduction: Causal Probabilities in Reinforcement Learning

The goal of decision-making in a Markov Decision Process (MDP) is to intervene in the environment to maximize the agent's cumulative reward. A key insight of causal decision theory is that the impact of an action should be estimated as a *causal effect*, not a correlation. Visits to the doctor correlate with illnesses, but avoiding seeing a doctor does not make a patient healthier (Pearl, 2000, Ch.4.1.1). Several causality researchers have therefore argued that reinforcement learning can benefit from adopting causal models to predict the effect of actions. This article is directed towards reinforcement learning researchers who want to explore the use of causal models. We provide conceptual and theoretical foundations to facilitate the adoption of causal models by reinforcement learning researchers. We use as much as possible terminology, notation, and examples from reinforcement learning. A running example gives explicit computations that illustrate causal concepts. This paper can therefore serve as a short tutorial on causal modeling for RL

Table 1: A four-level causal hierarchy, which refines Pearl's three-level hierarchy Association-Intervention-Counterfactual (Pearl, 2000). *Our analysis shows that with executable behavioral policies, queries of the first three types can be computed from conditional probabilities.*

| Level | Notation | Typical Question | Example |
|---|---|---|---|
| Association | $P(R|\mathbf{S}, A)$ | What reward follows after an agent chooses $A$? | How often does a shot lead to a goal? |
| Intervention | $P(R|\mathbf{S}, do(A))$ | If I choose $A$, what will my reward be? | If I take a shot, will I score a goal? |
| What-if Counterfactual | $P(R_A|\mathbf{S}, B)$ | What if I had chosen $A$ instead of $B$? | What if I had taken a shot instead of making a pass? |
| Hindsight Counterfactual | $P(R_A|\mathbf{S}, B, R)$ | How would my reward change if I had chosen $A$ instead of $B$? | I failed to score. What if I had taken a shot instead of making a pass? |

researchers. An excellent long tutorial is provided by Bareinboim (2020), a recent survey by Deng et al. (2023), and a book-length treatment by Bareinboim et al. (2025).

Conditional probabilities measure the strength of associations or correlations, but not necessarily the causal effect of an action. Using Pearl's do operator, the **causal effect** of setting variable $A$ to the value $a$ given evidence covariates $\mathbf{X}$ can be written as a conditional probability of the form $P(Y|do(A = a), \mathbf{X} = \mathbf{x}))$. (The formal semantics for the *do* operator is defined in Section 2 below.) Since causal effects are based on interventions, we refer them also as **interventional probabilities**. In the medical visit example, the strong correlation means that $P(Illness|Visit)$ is high. However, making a person visit the doctor has no causal effect on their illness, so we have $P(Illness|Visit) >> P(Illness|do(Visit)) = P(Illness)$. The question we address is *under what conditions causal probabilities provide a different approach to reinforcement learning than conditional probabilities.*

## 1.1 Summary of Main Result

Our analysis follows a "ladder of causation" as described by (Pearl, 2000): A hierarchy of probabilistic statements that require causal reasoning of increasing complexity. The levels correspond to associations, interventions, and counterfactuals. Table 1 illustrates these concepts in the RL setting. A *formal semantics* for each type of probability can be defined in terms of a generative model that is based on a causal graph. Among counterfactuals, we distinguish between *what-if* queries and *hindsight* queries. A what-if query concerns the results of deviating from an action taken; an example from a sports domain would be "What if I had taken a shot instead of making a pass?". A hindsight query conditions on an observed outcome. An example of a hindsight query would be "I failed to score. What if I had taken a shot instead of making a pass?". *Each level of the causal hierarchy supports a different policy evaluation method based on different types of reward and transition probabilities.* For example, conditional reward is given by the conditional probability $P(R_{t+1}|A_t = a_t, S_t)$ while the interventional reward is given by the interventional probability $P(R_{t+1}|do(A_t = a_t), S_t)$, where $P(R_{t+1}|A_t = a_t, S_t)$ is the conditional probability of receiving reward $R$ at time $t + 1$ given action $A_t$ and state $S_t$ at time $t$.

We define four Bellman equations for Q-functions based on (i) standard conditional probabilities, (ii) interventional probabilities, (iii) what-if probabilities, and (iv) hindsight probabilities. Our main theoretical result states that *if the behavioral data-generating policy is executable by the learning agent, then conditional Q-values, interventional Q-values, and what-if Q-values are equivalent.* Thus under executability, causal Q-values can be correctly estimated from conditional probabilities, as is done in standard RL.

By "executable" we mean that the observation signal available to the learning agent comprises all observations used by the behavioral policy. Figure 1 illustrates the concept in a simple sports setting (such as hockey

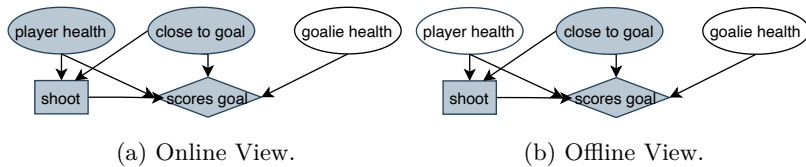

(a) Online View.  (b) Offline View.

Figure 1: Causal Graphs for a sports scenario like hockey or soccer to illustrate executability. We follow the conventions of influence diagrams to distinguish state variables, actions, and rewards. All variables are binary. Variables observed by the learning agent are gray, latent variables white. Whether a player takes a shot depends on their location and whether they are injured. Likewise, the chance of their shot leading to a goal depends on their location and health. Thus Player Health is a common cause of the action and reward. Figure 1a: In the online setting, the athlete learns from their own experience, which includes their health. Therefore a behavioral policy that depends on the agent's health is executable. Figure 1b: In the offline setting, the learner is different from the athlete, for example a coach, and does not observe the health of the behavioral agent. Therefore a behavioral policy that depends on the agent's health is not executable. Player health is an unobserved confounder of action and reward.

| RL setting | Executable? | what-if = interventional = conditional Q-values | Reason |
|---|---|---|---|
| On-policy | ✓ | ✓ | Learned policy = behavioral policy |
| Online exploration | ✓ | ✓ | Learner executes behavioral policy |
| Complete Observability | ✓ | ✓ | Learner's observation space = behavioral agent's observation space |
| Off-policy and Offline learning and partial observability | x | x | Behavioral policy relies on observations that are not observable to the learner |

Table 2: Reinforcement learning settings in which we can expect conditional probabilities to be equivalent to interventional probabilities and to what-if counterfactual probabilities. Online exploration refers to a setting in which the learning agent executes a behavioral policy to explore the environment. Hindsight counterfactuals cannot be reduced to conditional probabilities in any setting.

or soccer). Player health is a common cause of the player's decision (e.g., to shoot) and the player's success (score a goal). In the first-person online setting, the athlete is aware of their own health, and can execute a policy that depends on the athlete's health (e.g., "shoot only if healthy"). In the third-person offline setting, an observer such as a coach, does not have access to the athlete's health. They can execute only policies that depend on whether a player is close to the goal, but not those that depend on whether the player is healthy.

We show by example that hindsight Q-values differ from conditional Q-values even under executability. The fundamental reason is that *future* information provides extra information about *current* latent states over and above the past history. The special power of hindsight has been noted in previous RL work, such as the well-known hindsight experience replay approach (Andrychowicz et al., 2017), hindsight credit assignment (Harutyunyan et al., 2019), and data augmentation using hindsight (Sun et al., 2024).

## 1.2 Implications for RL settings

Next we discuss which RL settings satisfy the executability condition. Our overall conclusion, illustrated in Table 2, is that *what-if and interventional probabilities differ from conditional probabilities only in the offline off-policy RL setting with partial observability* (the POOO setting).

**RL settings with executable behavioral policies.** Common RL settings with behavioral policies that are executable by the learning agent include on-policy learning and online exploration, where the learning

agent uses a behavioral policy to explore the environment (e.g., $\epsilon$-greedy learning); see Appendix Figure 6 for illustration. In *on-policy* learning the behavioral and the learned policy coincide so they clearly share the same observation signal. Accordingly, Schölkopf et al. (2021) note that "[Reinforcement learning] sometimes effectively directly estimates do-probabilities. E.g., on-policy learning estimates do-probabilities for the interventions specified by the policy". Our work formalizes their observation that the on-policy setting is sufficient for conditional and interventional probabilities to coincide, and goes beyond it to consider other RL settings, such as the following.

In *online exploration*, the learning agent directly interacts with its environment to generate data and learn from their *own* experience exploring the environment (see Figure 6). For example a video game playing system can execute actions in the game and observe their effects (Mnih et al., 2015). Since the behavioral policy is executed by the learning agent to explore the environment, it must be executable. In the video game example, if the game playing agent employs an exploration policy, the policy must depend only on the information available in the video game.

Another sufficient condition for executability is *complete observability*, where the environment is completely observable for both the behavioral and the learning agent. For instance, the first phase of training the AlphaGo system was based on an offline dataset of games by Go masters (Silver et al., 2016). Go is a completely observable board game with no hidden information. Under complete observability, the learning agent has access to the same observations as the behavioral agent, as required for executability.

The Go example also illustrates how imitation learning can satisfy the executability condition if both the demonstrator and the learning agent (trainee), who aims to imitate them, share the same observation space. For an illustration in online imitation learning, in the DAgger approach (Ross et al., 2011), the trainee explores the state space using a mixture of the demonstrator's policy and the current learned policy. Using a mixture implies that both the expert and the trainee's policies operate with the same input observations.

**RL settings without executable behavioral policies.** When the behavioral policy is not executable by the learning agent, the learning agent may receive a different observation signal than the behavioral agent, as shown in Figure 2b. As we show in worked-out examples below, *without executability conditional probabilities may be confounded by the learning agent's missing information, and provide misleading estimates of expected rewards.* In the example of Figure 1b, Player Health is not observable by the learning agent, and thus confounds the decisions of the behavioral agent (i.e., whether the athlete shoots) with the rewards obtained by the behavioral agent (i.e., whether the athlete scores).

In the offline setting, the learning agent cannot directly interact with its environment, and instead relies on an offline dataset, which may not record all the observations available to the behavioral agent; see Figure 2. Several algorithms have been developed for imitation learning in a non-executable setting where the expert has access to more observations than the trainee. Ha et al. (2024) refer to this type of setting as privileged learning, whereas Warrington et al. (2021) use the term asymmetric imitation learning.

All results described so far hold even in the infinite sample setting with perfect probability estimates. With limited samples, confounding can become an issue even when the behavioral agent and the learning agent are one and the same. This is due to the implicit partial observability phenomenon described by Ghosh et al. (2021): when generalizing a policy from training contexts to a new test-time context, the agent may have less information at test time than at training time.[1] An example given by Ghosh et al. is navigating to the location of a target object using a map. The training data may allow the agent to train an image classifier with near-perfect accuracy to extract the location, whereas when tested in a new context, the image classifier may provide insufficiently accurate information. During training time, the output of the image classifier is a common cause of the agent's actions (route chosen) and the reward (target found). Ghosh et al. suggest that because of limited samples, at test time the agent has only partial observability of the image information, which in contrast is perfectly observable during training. In our terminology, the test time agent is in a POOO setting, and policy estimates based on conditional probabilities from the training data will be confounded.

---

[1]We owe this point to an anonymous reviewer.

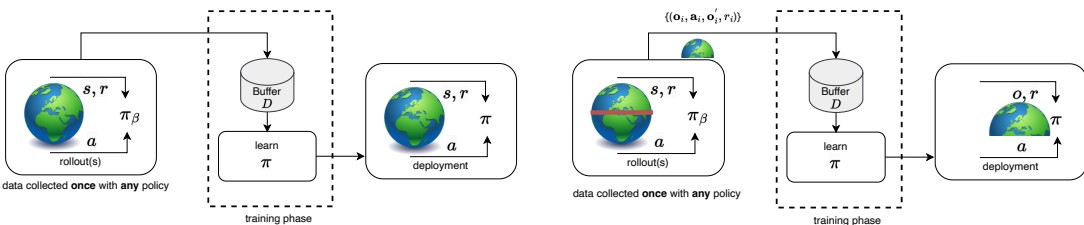

(a) Offline RL: executable behavioral policies.     (b) Offline RL: non-executable behavioral policies.

Figure 2: **Offline** RL-learning employs a dataset $D$ collected by some (potentially unknown) behavior policy $\pi_\beta$. The dataset is collected once, and is not altered during training. The training process does not interact with the environment directly, and the policy is only deployed after being fully trained. Figure 2a (Levine et al., 2020): Offline RL with executability where the behavioral policy $\pi_\beta$ and the learned policy $\pi$ are based on the same observation signal. Figure 2b: Offline RL without executability where the behavior policy $\pi_\beta$ has access to more observations than the learned policy $\pi$.

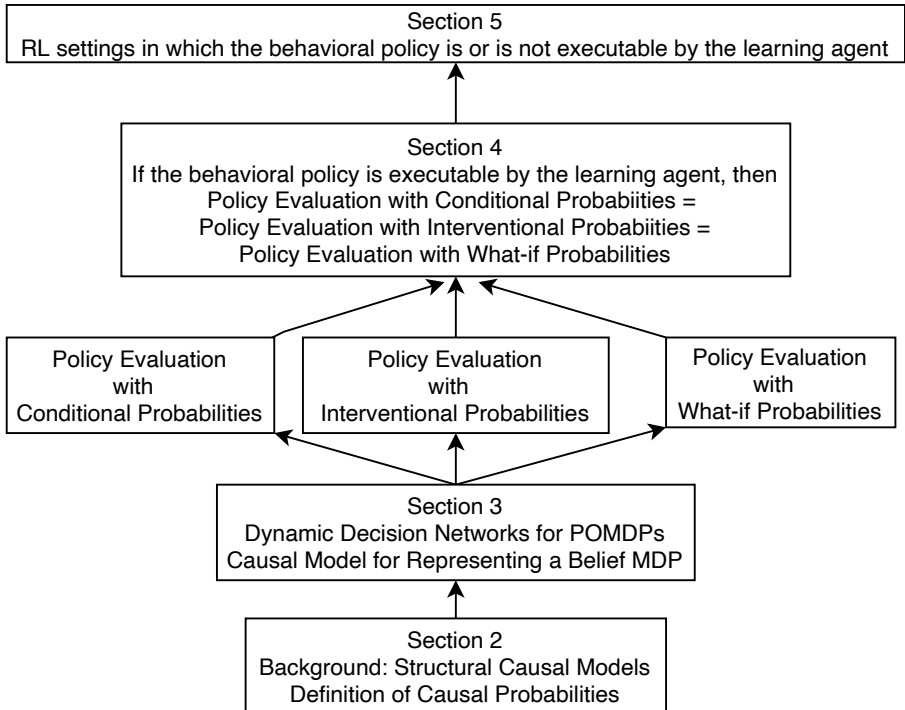

Figure 3: Topics by paper section and connections between them. POMDP = Partially Observable Markov Decision Process.

**Paper Overview**    Figure 3 illustrates our paper organization. Section 2 gives background on causal models and how they define a precise semantics for interventional and counterfactual probabilities. Since latent confounders are the key difference between causation and correlation, partially observable Markov Decision Processes (POMDPs) are the relevant RL model for understanding the importance of causal reasoning. To port causal concepts to RL, we therefore require a type of causal model that can represent POMDPs. Section 3 describes a new variant of *dynamic decision network* (DDN) (Russell & Norvig, 2010, Ch.17) for representing POMDPs. In a DDN, an environment state is represented by an assignment of values to state variables. A DDN models causes of both the agent's actions and the environment's responses.

Section 4 shows how a DDN supports model-based policy evaluation in three different ways: based on conditional probabilities, interventional probabilities, or what-if probabilities. Our main formal proposition

| Notation | Meaning |
|---|---|
| $\boldsymbol{X}$ | Generic Random Variables. Nodes in a Causal Graph. $\boldsymbol{X} = \boldsymbol{O} \cup \boldsymbol{Z}$ |
| $\boldsymbol{O}$ | Observed Variables. |
| $\boldsymbol{Z}$ | Latent Variables |
| $\boldsymbol{U}$ | Source Variables in a causal graph. |
| $\underline{\boldsymbol{X}} = \boldsymbol{X} - \boldsymbol{U}$ | Non-source variables with positive indegree |
| $G$ | Directed Acyclic Graph (DAG); causal graph |
| $Pa_X$ resp. $Pa_i$ | Parents/direct causes of variable $X$ resp. $X_i$ |
| $C$ | Probabilistic Structural Causal Model |
| $D$ | Dynamic Decision Network |
| $f_X$ resp. $f_i$ | Deterministic local function for variable $X$ resp. $X_i$ |
| $S$ | State Space in POMDP |
| $\boldsymbol{S}$ | State Variables in factored POMDP; $\boldsymbol{S} = \boldsymbol{O} \cup \boldsymbol{Z} - \{A, R\}$ |
| $A$ | Intervention Target; Decision Variable in Influence Diagram; Action in POMDP |
| $do(A = \hat{a})$ | Selecting action $\hat{a}$ as an intervention |
| $R$ | Reward |
| $b(\boldsymbol{Z})$ | Belief State |
| $\pi$ | Policy |
| $\pi(a|\langle \boldsymbol{o}, b \rangle)$ | Probability of action given current observation and belief state |
| $Q^\pi(\langle \boldsymbol{o}, b \rangle, a)$ | Expected Return given current observation, belief state, and action |
| $Q^\pi(\langle \boldsymbol{o}, b \rangle, do(\hat{a}))$ | Expected Return given current observation, belief state, and action/intervention |

Table 3: Notation used in this paper

states that when the behavioral policy is executable by the learning agent, policy evaluation with conditional probabilities is equivalent to policy evaluation with interventional/what-if probabilities and what-if counterfactuals. *Our results do not assume that the learning agent is given a true causal model of the behavioral policy and the environmental process.* As long as the learning agent's observation signal includes the causes of the behavioral agent's actions, conditional probabilities inferred from the action data are equivalent to interventional and what-if probabilities, no matter what the true causal model is.

Section 5 shows how the causal policy evaluation theorem can be used to understand which RL settings do and do not satisfy executability, and therefore do or do not require causal policy evaluation. Our final section utilizes the concepts and insights developed in the main body to review related work on causal models in RL, and describe directions for future research from the perspective of confounding and executability.

## 2 Background: Structural Causal Models and Counterfactuals

**Notation**  Table 3 summarizes the notation used in this paper, and previews the concepts introduced in the remainder.

Our goal in this section is to define the class of structural causal models, which give a precise semantics for interventional and counterfactual probabilities, collectively called **causal probabilities**.

### 2.1 Structural Causal Models

Structural causal models (SCMs) combine a causal graph with latent variables and deterministic local functions that map parent values to child values (Pearl, 2000). SCMs are compatible with deep learning, in that the local functions are decoders in the sense of deep generative models. Deep generative models with latent variables are therefore a powerful architecture for implementing and learning models for counterfactual reasoning (Geffner et al., 2022). Our presentation of SCMs follows those of Pearl (2000) and of Schölkopf et al. (2021).

A **structural causal model** (SCM) is a pair $\mathcal{S} = \langle G, \boldsymbol{F} \rangle$ meeting the following conditions.

- $G$ is a DAG over random variables $\boldsymbol{X}$. Let $\boldsymbol{U}$ be the set of **source nodes** with indegree 0 in $G$. We write $\underline{\boldsymbol{X}} = \boldsymbol{X} - \boldsymbol{U}$ for the set of non-source variables.

- $\boldsymbol{F} = \{f_1, \ldots, f_n\}$ is a set of local functions such that each $f_i$ deterministically maps the parents of non-source variable $X_i \in \underline{\boldsymbol{X}}$ to a value of $X_i$. The local functions are often written in the form of a **structural equation**:

$$X_i = f_i(pa_i). \tag{1}$$

The source variables are also called *background variables* or *exogenous variables*. They can be, and often are, latent variables. For example, in a linear structural equation

$$f(X, \varepsilon) \equiv Y = aX + b + \varepsilon$$

the parent $X$ of $Y$ represents an observed cause and the noise term $\varepsilon$ can be modeled as a latent parent summarizing unobserved causal influences. Given an assignment $\boldsymbol{U} = \boldsymbol{u}$ of values to the source variables, we can compute values for the non-source variables $\underline{\boldsymbol{X}}$ by using the local functions to assign values to the children of the source nodes, then to the children of the children, etc. This recursive evaluation procedure defines a **solution function**

$$F^{\mathcal{S}}(\boldsymbol{u}) = \underline{\boldsymbol{x}} \tag{2}$$

where $\underline{\boldsymbol{x}}$ is an assignment of values to the non-source nodes $\underline{\boldsymbol{X}}$. A **probabilistic SCM** is a pair $C = (\mathcal{S}, b)$ where $b$ is a joint **prior distribution** over the source variables $\boldsymbol{U}$ (Pearl, 2000, Eq.7.2). A probabilistic SCM $C$ defines a **joint distribution** over the variables $\boldsymbol{X} = \boldsymbol{U} \cup \underline{\boldsymbol{X}}$:

$$P^C(\underline{\boldsymbol{X}} = \underline{\boldsymbol{x}}, \boldsymbol{U} = \boldsymbol{u}) = P^C(\underline{\boldsymbol{X}} = \underline{\boldsymbol{x}} | \boldsymbol{U} = \boldsymbol{u}) \times b(\boldsymbol{U} = \boldsymbol{u}) \tag{3}$$

$$b(\boldsymbol{U} = \boldsymbol{u}) = \prod_{U \in \boldsymbol{U}} b(U = u) \tag{4}$$

$$P^C(\underline{\boldsymbol{X}} = \underline{\boldsymbol{x}} | \boldsymbol{U} = \boldsymbol{u}) = \begin{cases} 1, & \text{if } F^{\mathcal{S}}(\boldsymbol{u}) = \underline{\boldsymbol{x}} \\ 0, & \text{otherwise} \end{cases} \tag{5}$$

where $F^{\mathcal{S}}(\boldsymbol{u})$ is the solution function (Equation (2).) Equation (4) says that the unconditional prior distribution over source variables factors into individual priors over each source variable. Equation (5) applies the deterministic solution function as a deterministic decoder that maps the source variables to a unique assignment for non-source variables. Section 3.2 below gives an example of an SCM.

As usual with generative models, a probabilistic SCM defines a mixture distribution over observed variables through marginalization

$$P_{\mathcal{S}}(x_1, \ldots, x_n) = \sum_{\boldsymbol{z}: F_{\mathcal{S}}(\boldsymbol{z}) = (x_1, \ldots, x_n)} P(\boldsymbol{z}) \tag{6}$$

## 2.2 Interventional Probabilities

An **interventional distribution** (Pearl, 2000, Dfn. 7.1.3) is denoted as $P(\boldsymbol{Y} | do(A = \hat{a}))$, where $\boldsymbol{Y}$ and $A$ comprise observed variables. It represents the *causal effect* (Pearl, 2000, Dfn.3.2.1.) of intervening in the environment—the effect on variables $\boldsymbol{Y}$ of setting a variable $A$ to a possible value $\hat{a}$. In graphical terms, the effect of an intervention is computed by removing links pointing into $A$, fixing the values of $X$ variables to the assignment $\hat{a}$, then computing the joint probability of $\boldsymbol{Y}$ in the truncated model. Removing the parents of a variable represents that an intervention removes the causal influence of the parents. For example, if the shooting decision of a player is caused by their health and their distance to the goal, then if we intervene to make the player shoot, their decision is determined and no longer depends on other variables.

*Remark on notation.* The $\hat{a}$ notation does not indicate a quantity estimated from data, but an intervention. We sometimes use the syntactic sugar $\hat{A}$ to highlight a context where $A$ is intervened upon. In our applications to RL, we consider intervening only on a special variable $A$ that represents the agent's actions. However, the intervention semantics is well-defined for manipulating *any* random variable in a causal model, not only a designated special action/decision variable.

Formally, an intervention that assigns value $\hat{a}$ to variable $A$ is represented by the truncated **submodel** $\mathcal{S}_{\hat{a}} = \langle G_A, \boldsymbol{F}_{\hat{a}} \rangle$, which is the causal model where $G_A$ contains all edges in $G$ except those pointing into variable $A$, and $\boldsymbol{F}_{\hat{a}} = \{f_i : X_i \neq A\} \cup \{A = \hat{a}\}$. Here $\{f_i : X_i \neq A\}$ is the set of all local functions for unmanipulated variables, and $A = \hat{a}$ is the constant function that assigns variable $A$ its manipulated value. Similarly let $b_{\hat{a}}$ be the prior distribution over source node variables that assigns probability 1 to $\hat{a}$ and agrees with $b$ on all other variables. Formally, $b_{\hat{a}}(A = \hat{a}) = 1$, and $b_{\hat{a}}(U = u) = b(U = u)$ for $U \neq A$. We compute the intervention distribution as the joint probability in the truncated submodel:

$$P^{(\mathcal{S},b)}_{do(A=\hat{a})}(\boldsymbol{X} = \boldsymbol{x}) = P^{(\mathcal{S}_{\hat{a}}, b_{\hat{a}})}(\boldsymbol{X} = \boldsymbol{x}) \tag{7}$$

where $P^{(\mathcal{S},b)}_{do(A=\hat{a})}$ is the interventional distribution of the causal model $(\mathcal{S}, b)$ and $P^{(\mathcal{S}_{\hat{a}}, b_{\hat{a}})}$ is the joint distribution Equation (6) of the truncated submodel $(\mathcal{S}_{\hat{a}}, b_{\hat{a}})$ defined by the intervention. Appendix Figure 11 illustrates the truncation semantics. We next show how the interventional distribution can be used to define a formal semantics for counterfactuals.

### 2.3 Counterfactual Probabilities

A **counterfactual probability** $P(\boldsymbol{Y}_{\hat{a}} = \boldsymbol{y}' | \boldsymbol{X} = \boldsymbol{x}, A = a, \boldsymbol{Y} = \boldsymbol{y})$ can be read as follows: "Given that we observed action $A = a$, and state variables $\boldsymbol{X} = \boldsymbol{x}$, followed by outcome $\boldsymbol{Y} = \boldsymbol{y}$, what is the probability of obtaining an alternative outcome $\boldsymbol{y}'$, if we were to instead select the action $\hat{a}$ as an intervention?" Here $\boldsymbol{Y}_{\hat{a}}$ is a list of **potential outcome** random variables, distinct from the actual outcomes $\boldsymbol{Y}$, and $\boldsymbol{X}, A, \boldsymbol{Y}$ are disjoint. For a given probabilistic SCM $C = (\mathcal{S}, b)$, we can compute the counterfactual probability as follows (Pearl, 2000, Th.7.1.7):

**Abduction/Posterior Update** Condition on the observations $\boldsymbol{X} = \boldsymbol{x}, A = a, \boldsymbol{Y} = \boldsymbol{y}$ to compute a source variable posterior
$$b' \equiv b(\boldsymbol{U} | \boldsymbol{X} = \boldsymbol{x}, A = a, \boldsymbol{Y} = \boldsymbol{y}).$$

**Intervention** Apply the intervention $do(A = \hat{a})$ to compute the submodel $\mathcal{S}_{\hat{a}}$ and the SCM $C' = (\mathcal{S}_{\hat{a}}, b'_{\hat{a}})$.

**Prediction** Return the conditional probability $P(\boldsymbol{Y} = \boldsymbol{y}' | \boldsymbol{X} = \boldsymbol{x})$ computed in the updated SCM:

$$P^C(\boldsymbol{Y}_{\hat{a}} = \boldsymbol{y}' | \boldsymbol{X} = \boldsymbol{x}, A = a, \boldsymbol{Y} = \boldsymbol{y}) = P^{C'}(\boldsymbol{Y} = \boldsymbol{y}' | \boldsymbol{X} = \boldsymbol{x}) \tag{8}$$

A posterior update is often called abduction in causal modeling. Through their posterior $b'$, the source variables carry information from the observed configuration $\boldsymbol{X} = \boldsymbol{x}, A = a, \boldsymbol{Y} = \boldsymbol{y}$ to the counterfactual configuration where $\boldsymbol{X} = \boldsymbol{x}, A = \hat{a}, \boldsymbol{Y} = \boldsymbol{y}'$. Appendix Figure 11 illustrates a counterfactual computation. Counterfactual probabilities form a natural hierarchy as follows.

- We refer to the most general counterfactual of the form $P(\boldsymbol{Y}_{\hat{a}} | \boldsymbol{X}, A, \boldsymbol{Y})$ as a **hindsight** counterfactual query because it specifies the actual outcomes $\boldsymbol{Y}$.

- If the actual outcomes $\boldsymbol{Y}$ are *not* included in the evidence, we have a **what-if** counterfactual query $P(\boldsymbol{Y}_{\hat{a}} | \boldsymbol{X}, A)$ that asks what the likely outcome is after deviating from the actual choice $A$. For what-if queries, we use the causal effect notation $P(\boldsymbol{Y} | \boldsymbol{X}, A, do(A = \hat{a})) \equiv P(\boldsymbol{Y}_{\hat{a}} | \boldsymbol{X}, A)$.

- If neither an observed outcome nor an observed action are specified, a counterfactual probability reduces to an interventional probability $P(\boldsymbol{Y} | \boldsymbol{X}, do(A = \hat{a}))$.

- If neither an observed outcome nor an intervention are specified, a counterfactual probability reduces to a conditional probability $P(\boldsymbol{Y}|\boldsymbol{X}, A)$.

*Results that we prove for more complex queries immediately hold for simpler queries since they are special cases.* For instance, a theorem for what-if probabilities also covers interventional and conditional probabilities. In the next section we compare policy evaluation based on conditional and causal probabilitiies.

## 3   Dynamic Decision Networks for POMDPS

This section describes a structural causal model for representing POMDPs. The causal model supports a precise semantics for conditional/interventional/counterfactual probabilities relevant to RL, notably transition and reward probabilities.

POMDP theory and causal concepts share a common formal structure, despite differences in terminology for describing interventions and their effects. Table 4 shows translations between analogous concepts.

Table 4: Correspondence between Causal and RL terminology.

| Reinforcement Learning | Causality |
|:---:|:---:|
| action | treatment |
| reward | response |
| observed state $O$ | observed co-variates $\boldsymbol{O}$ |
| state $S$ | co-variates $\boldsymbol{O} \cup \boldsymbol{Z}$ |
| belief state $b(S)$ | latent variable distribution $b(\boldsymbol{Z})$ |
| belief state update $b(S|O)$ | abduction $b(\boldsymbol{Z}|\boldsymbol{O})$ |
| complete observability | causal sufficiency |

A key difference is that RL concepts are usually defined in terms of a single state $s$, whereas causal concepts are defined in terms of values for a list of variables. Using the terminology of Russell & Norvig (2010, Ch.2.4.7), RL uses an *atomic* environment representation, and causal models use a *factored* representation, where a state is represented as an assignment $\boldsymbol{S} = \boldsymbol{s}$ to a set of *state variables* $\boldsymbol{S}$. Graphical models for factored MDPs are known as *dynamic influence diagrams* (Polich & Gmytrasiewicz, 2007). The basic idea of a dynamic graphical model is to make a copy $\boldsymbol{X}'$ for the random variables in a causal graph, to represent successor variables. The dynamic causal graph is then a causal graph over the current and successor variables (i.e., over $\boldsymbol{X} \cup \boldsymbol{X}'$), such that there are no edges from the successor variables to the current variables.

### 3.1   Definition of Dynamic Decision Network

Since unobserved confounders are the key difference between causation and correlation, the need for causal reasoning arises in partially observable MDPs (POMDPs). In order to apply MDP techniques to POMDPs, they are often transformed into a *belief MDP*. Following Russell & Norvig (2010), we use the term **dynamic decision network** (DDN) for a dynamic influence diagram that represents a belief MDP. Appendix B reviews background on factored POMDPs without reference to causal graphs, including the transformation to belief MDPs. We introduce a new DDN variant for belief MDPs.

**Definition 1.** *A **dynamic decision network** (DDN) D for state variables $\boldsymbol{S}$ comprises the following random variables.*

1. *Current time slice: $\boldsymbol{X} = \boldsymbol{S} \cup \{A\} \cup \{R\} \cup \{B\}$*

2. *Next time slice: $\boldsymbol{X}' = \boldsymbol{S}' \cup \{A'\} \cup \{R'\} \cup \{B'\}$.*

*where B resp. B' represent the agent's current resp. successor beliefs.*

*A DDN D satisfies the following causal assumptions.*

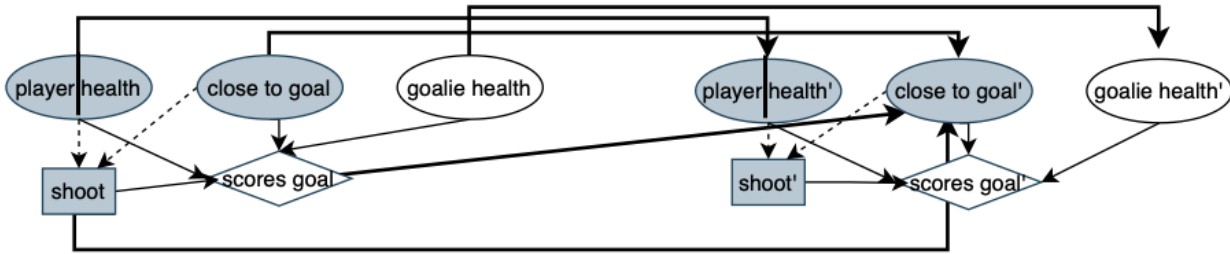

Figure 4: A dynamic decision network graph for our sports scenario. The reward model is indicated by thin lines, and the state transition model by thick lines. Dashed lines represent the agent's policy. (The agent's belief state is not shown, see text) The line through the Player Health node indicates that in *the online version (a)*, Player Health is observable, and in the *offline version (b), Player Health is not observable.*

1. *There are no edges from $\boldsymbol{X'}$ to $\boldsymbol{X}$.*

2. *There are no edges from $R$ to nodes in $\boldsymbol{X}$, and no edges from $R'$ to nodes in $\boldsymbol{X'}$.*

3. *There are no edges from $A$ to nodes in $\boldsymbol{X} - \{R\}$, and no edges from $A'$ to nodes in $\boldsymbol{X'} - \{R'\}$.*

These assumptions state that (1) causal relationships respect the temporal ordering, (2) rewards may causally depend on the current state and action, but not vice versa, (3) actions may causally depend on the current state, but not vice versa, and not on the current reward.

De Haan et al. (2019) argue that Assumption 3 can fail if the temporal resolution of events is low enough that an action is assigned the same discrete time index as the effect of an action, resulting in "causal confusion". We discuss this scenario further under related work. Our main conclusions do not depend on Assumption 3, but we use it to simplify formal arguments. Russell & Norvig (2010, Ch.17.4) assume that in a DDN, the only parent of an action is the agent's belief state $B$. However, this assumption does not allow us to represent state variables as direct causes of actions (e.g., the health of a player causes them to shoot).

A fully specified influence diagram, or DDN, defines a joint distribution $P^D(\boldsymbol{X}, \boldsymbol{X'})$ over both time slices, as explained in Section 2. A DDN therefore specifies an environment MDP as follows.

- $P^D(\boldsymbol{S})$ is the initial state distribution.

- $P^D(\boldsymbol{S'}|\boldsymbol{S}, A)$ is the state transition model.

- $P^D(R|\boldsymbol{S}, A)$ is the reward model.

We next give an extended example of a DDN for our sports scenario, which illustrates how a DDN supports causal probabilities for RL.

### 3.2 Dynamic Decision Network Example

Figure 4 shows the causal graph for a dynamic version of our sports example.

**Structural Equations.** The graph is parametrized by the structural equations of Table 5, which can be read as follows.

For the reward model, a player scores if and only if they shoot, are healthy, close to the goal, and the goalie is not healthy. In the online model Figure 4(i), the player shoots if and only if they are healthy and close to the goal. We treat the player and goalie health as persistent time-invariant features (Lu et al., 2018). Finally for the location dynamics, we make the simplifying assumption that the attacking team stays close to the goal if and only if they maintain possession. Any state with $CG = 0$ is therefore an absorbing state: $P^D(CG' = 0|CG = 0) = 1$. (2) If the attacking team is close to the goal, our scenario works as follows.

| Variable | Equation |
|---|---|
| $Shoot(SH)$ | $SH = PH \cdot CG$ |
| $Scores(SC)$ | $SC = PH \cdot CG \cdot SH \cdot (1 - GH)$ |
| $PlayerHealth'(PH')$ | $PH' = PH$ |
| $GoalieHealth'(GH')$ | $GH' = GH$ |
| $ClosetoGoal'(CG')$ | $CG' = \begin{cases} 1, & \text{if } CG = 0 \text{ or } SH = 1 \\ Z_{CG'}, & \text{if } CG = 1 \text{ and } SH = 0 \end{cases}$ |

Table 5: A set of structural equations for the causal graph of Figure 4. $Z_{CG'}$ denotes a uniformly distributed binary noise variable for $CG'$ (not shown in Figure 4).

| | Online fig. 4(a) | Offline fig. 4(b) |
|---|---|---|
| Conditional | $P(SC = 1 \mid PH = 1, CG = 1, SH = 1) = 1/2$ | $P(SC = 1 \mid CG = 1, SH = 1) = 1/2$ |
| Intervention | $P(SC = 1 \mid PH = 1, CG = 1, do(SH = 1)) = 1/2$ | $P(SC = 1 \mid CG = 1, do(SH = 1)) = 1/4$ |
| What-if | $P(SC = 1 \mid PH = 1, CG = 1, SH = 0, do(SH = 1)) = 1/2$ | $P(SC = 1 \mid PH = 1, CG = 1, SH = 0, do(SH = 1)) = 0$ |

Table 6: Scoring probabilities for the reward model in the soccer examples of Figure 4. Appendix F shows the steps for computing these values.

- If the player shoots, the team loses possession, either because they scored, or because the shot was blocked and the defending team took over (no rebounds): $P_E(CG' = 0 \mid SH = 1) = 1.)$.

- If the player does not shoot (e.g., they pass instead), there is a 50% change that the attacking team retains possession.

The source variable distribution is uniform:

$$1/2 = P^D(PH = 1) = P^D(CG = 1) = P^D(GH = 1). \tag{9}$$

Figure 4 does not feature a belief state variable $B$, which implies that *the agent's decision is Markovian* in that it depends only on the current state variables, but not the agent's current beliefs. Markovian policies are also assumed in the theory of confounded MDPs (Zhang & Bareinboim, 2016; Kausik et al., 2024; Bruns-Smith, 2021). Our theorems and analysis address the general case of non-Markovian policies that can depend on the agent's entire observation history. Russell & Norvig (2010, Ch.17.4) provide an example DDN with a continuous belief state.

**Example: Causal Reward Probabilities.** Table 6 shows the values of conditional and causal reward probabilities for our example SCM. In the offline model where Player Health is not observed, it induces a non-causal correlation between action and reward, which leads to conditional probability of scoring of $1/2$, which is higher than the intervention probability $1/4$. However, if the common cause between action and reward is observed, as in the online model, conditioning on it eliminates the non-causal correlation, and the conditional and causal probabilities agree. The next section shows that this is an instance of a general pattern that extends to Q-values: When the causes of the agent's action are observable, conditional, interventional, and what-if policy values agree.

## 4    Policy Evaluation for Belief MDPs

The **policy evaluation task** is to compute the value function for a **policy** $\pi$ in a given environment. In a typical RL setting, we evaluate a policy $\pi$ learned from data generated by a **behavioral** policy $\pi_\beta$; see Figure 2.[2]

---

[2]The learned policy is called the estimation policy in (Sutton & Barto, 1998, Ch.5.6), the target policy by Wan et al. (2021), and the evaluation policy by Bruns-Smith (2021).

In a factored POMDP, the agent's decisions depend not only on their current belief state, but also on their current observation. Accordingly, an **epistemic state** $\langle \boldsymbol{o}, b \rangle$ comprises a current observation $\boldsymbol{o}$ and a current belief state. We use the tuple notation $\langle \rangle$ to make longer formulas easier to parse, and to emphasize the analogy between epistemic states and MDP states in a traditional atomic representation. A **policy** maps an epistemic state to a distribution over actions:

$$\pi : \boldsymbol{O}^\pi \times B^\pi \to \Delta(A)$$

where $\boldsymbol{O}^\pi$ represents the **observation space** of the agent executing policy $\pi$ and $B^\pi$ the space of belief states, i.e. distributions $\Delta(\boldsymbol{Z}^\pi)$ over the unobserved variables. We also use the conditional probability notation $\pi(a|\langle \boldsymbol{o}, b \rangle)$.

Since a DDN defines an environment process, we can use it to perform *model-based evaluation of a policy*, by computing the required reward and transition probabilities from the DDN. We focus on what-if reward and transition probabilities, since they cover interventional and observational probabilities as a special case. In RL based on what-if counterfactual decisions, the learning agent seeks to learn an optimal policy that may deviate from the actual decisions taken. Following Pearl's suggestion in a similar context (Pearl, 2000, Ch.4.1.1), we refer to observed decisions by the behavioral agent as "acts", with associated random variable $A$ and active decisions by the learning agent as "actions", with associated random variable $\hat{A}$.

### 4.1 Bellman Equation for Causal Q-functions

Given a policy $\pi$, the **Bellman equation for the what-if Q-function** is as follows.

$$Q^{\pi,D}(\langle \boldsymbol{o}, a, b \rangle, do(\hat{a})) = R^D(\langle \boldsymbol{o}, a, b \rangle, do(\hat{a})) + \gamma \sum_{\boldsymbol{o}'} P^D(\boldsymbol{o}'|\langle \boldsymbol{o}, a, b \rangle, do(\hat{a})) V^{\pi,D}(\boldsymbol{o}', a, b') \tag{10}$$

$$V^{\pi,D}(\langle \boldsymbol{o}, a, b \rangle) = \sum_{\hat{a}} \pi(\hat{a}|\langle \boldsymbol{o}, a, b \rangle) Q^{\pi,D}(\langle \boldsymbol{o}, a, b \rangle, do(\hat{a})) \tag{11}$$

$$R^D(\langle \boldsymbol{o}, a, b \rangle, do(\hat{a})) = E_{\boldsymbol{z} \sim b(\boldsymbol{z}|do(\hat{a}))} \sum_r r \cdot P^D(R = r|\boldsymbol{z}, \boldsymbol{o}, a, do(\hat{a}))$$

where $\gamma \in (0, 1]$ is a discount factor. In the equation shown, the epistemic states $\langle \boldsymbol{o}, a, b \rangle$ are defined by the policy's observation space $\boldsymbol{O}^\pi$. The $P^D$ terms are defined by the DDN as shown in Section 3. As usual the state value $V$ is the expectation of $Q$-values over actions the agent may take in the state (eq. (11)). Equation (10) finds the current $Q$-value by computing the current expected reward and the expected value of the state reached after the next observation $\boldsymbol{o}'$.

According to recurrent Equation (10), given a new observation $\boldsymbol{o}'$ and the current action $a$, the expected policy value can be computed in two steps:

**Posterior Update** Compute the new belief state $b'$ by conditioning the current beliefs $b$ on observations $\boldsymbol{o}, \boldsymbol{o}'$ and action $a$. The posterior belief $b'$ can be computed by the well-known belief update formula, which we derive for a factored POMDP in Appendix B.2 (Equation (16)).

**Prediction** Estimate the expected return $V^\pi(\boldsymbol{o}', b')$ given the new observation $\boldsymbol{o}'$ and new belief state $b'$.

Updating a posterior to predict the outcome of an action $a$ is a key part of the formal semantics of counterfactuals that we presented in Section 2. Appendix G illustrates how the different Bellman equations can be used to evaluate a policy. Bellman equations for the interventional Q-function $Q^{\pi,D}(\langle \boldsymbol{o}, b \rangle, do(\hat{a}))$ and the traditional observational Q-function $Q^{\pi,D}(\langle \boldsymbol{o}, b \rangle, a)$ are derived as special cases of Equation (10). Appendix C writes them out and shows how the interventional and what-if Q-functions are straightforwardly obtained from the standard observational Bellman equation. Policy evaluation based on the interventional Q-function has been studied in previous work (Wang et al., 2021; Zhang & Bareinboim, 2020).

### 4.2 Causal versus Conditional Policy Evaluation

Our main policy evaluation result states that if the causes of the action variables are observable, then what-if Q-values, interventional Q-values, and conditional Q-values are equivalent. A technical issue is that in an SCM, each variable may have an unobserved noise variable as a cause. Instead of complete observability, the causal modeling literature therefore utilizes a more general *causal sufficiency* condition that allows a variable to have a latent cause, but not a *shared* latent cause (Spirtes et al., 2000), that is, a confounder. Formally we say that source variable $U$ is a **noise variable** for $X$ if $U$ is independent of every variable except $X$. In Figure 4, goalie health is a noise variable for scoring. A set of observed variables $\boldsymbol{O}$ is **causally sufficient** for variable $X$ in a causal graph $G$ if every latent parent of $X$ is a noise variable for $X$. An SCM $C$ is **action sufficient** if its graph is causally sufficient for action variable $A$.[3] The next proposition states our main result.

**Theorem 1.** *Suppose that an observation signal $\boldsymbol{O}$ is action sufficient in a dynamic probabilistic SCM $D$. Then for every epistemic state, observed act $a$, and action $\hat{a}$:*

$$Q^{\pi,D}(\langle \boldsymbol{o}, a, b\rangle, do(\hat{a})) = Q^{\pi,D}(\langle \boldsymbol{o}, b\rangle, do(\hat{a})) = Q^{\pi,D}(\langle \boldsymbol{o}, b\rangle, \hat{a}).$$

*That is, given that the direct causes of an act have been observed, the Q-value associated with an intervention $\hat{a}$ replacing the actual act $a$ equals the Q-value associated with conditioning on $\hat{a}$, independent of the act.*

The upshot is that *under action sufficiency, what-if and interventional Q-values can be evaluated using conditional probabilities*, as is done in traditional RL. The proof of Theorem 1 is given in Appendix D.1. An informal outline is as follows: We show as a general lemma in causal theory that if the parent values of a manipulated variable $A$ are given, what-if probabilities intervening on $A$ are equivalent to conditional probabilities observing $A$. This result can be used to show that what-if reward/transition probabilities are equivalent to observational reward/transition probabilities. Since the Bellman equations define Q-values recursively in terms of reward/transition probabilities, the equivalence of reward/transition probabilities implies the equivalence of the Q-functions.

**Remarks**  Although Theorem 1 is stated in terms of a causal model, it does *not* assume that the learning agent is given a true causal model of the behavioral policy and the environmental process. Rather, the import is that as long as the learning agent's observation signal includes the causes of the behavioral agent's actions, conditional probabilities inferred from the action data are equivalent to interventional probabilities, no matter what the true dynamic causal model is.

Theorem 1 does not cover hindsight Q-values. Indeed a remarkable feature of counterfactual hindsight Q-values is that *they can differ from conditional probabilities even under action-sufficiency.* Appendix E provides worked-out counterexamples.

**Examples**  Table 7 illustrates Theorem 1. The Q-values are similar to the reward values in Table 6, so we show the state values, which capture the sequential dynamics of Bellman's equation. In the on-line model

---

[3]Our concept of action-sufficiency differs from the notion introduced by Huang et al. (2022). For them, a latent state is action sufficient if it is powerful enough to support an optimal policy based on the latent state space. Our concept could be called "causal action sufficiency" to disambiguate.

|  | Online Figure 4 | Offline Figure 4 |
|---|---|---|
| Conditional | $V^{\pi,D}(PH=1, CG=1, b) = 1/2$ | $V^{\mu,D}(CG=1, b) = 1/3$ |
| Intervention | $V_{do}^{\pi,D}(PH=1, CG=1, b) = 1/2$ | $V_{do}^{\mu,D}(CG=1, b) = 1/6$ |
| What-if | $V_{do}^{\pi,D}(PH=1, CG=1, SH=0) = 1/2$ | $V_{do}^{\mu,D}(CG=1, b) = 0$ |

Table 7: Value functions in the soccer examples of Figure 4. In the online version of Figure 4, Player Health is observable, in the offline version it is not. The belief $b$ is the uniform prior over source variables. Appendix G shows the steps for computing the offline model values. The notation $V_{do}$ denotes a value function that averages over next step interventional Q-values (Equation (10)).

of Figure 4, where Player Health is observable, we evaluate the standard behavioral policy $\pi$ (shoot if and only if close to the goal and healthy). Since this policy shoots deterministically when close to the goal, the state values agree with the reward probabilities of Table 6.

For the offline model of Figure 4, where Player Health is not observable, we evaluate the **marginal** policy $\mu$ derived from the behavioral policy by averaging over latent states. The marginal policy is an important concept in offline policy evaluation (Kausik et al., 2024; Bruns-Smith, 2021). It can be viewed as a baseline form of behavioral cloning that estimates the agent's action probability from frequencies based on the observation signals. In our example, if the player is close to the goal with unknown health status, the probability that they take a shot is $1/2$. So we have the following standard marginal policy for our sports example:

$$P(SH = 1|CG) = \begin{cases} 1/2, & \text{if } CG = 1 \\ 0, & \text{otherwise} \end{cases} \tag{12}$$

Observe in Table 7 that in the non-executable offline setting, the marginal policy has different values depending on whether it is evaluated based on conditional, interventional, or what-if probabilities.

## 5 Action Sufficiency in RL Settings

This section connects the action sufficiency condition of Theorem 1 to policy executability and reviews the importance of our theoretical analysis for common RL settings.

A policy $\pi$ is **executable** if $\pi(a|\langle \boldsymbol{o}, b \rangle) = \pi(a|\langle \boldsymbol{o}, b \rangle, \boldsymbol{z})$ for any latent state variable $\boldsymbol{z}$ that is not a noise variable for $A$. Executability is equivalent to the act-state independence condition discussed by Gasse et al. (2021). Intuitively, executability guarantees that the policy observation space contains sufficient information for assigning probabilities to decisions in accordance with the policy.

Just as a DDN defines conditional probabilities that model an environmental process, it defines conditional probabilities that model a behavioral policy. We say that a DDN **matches a behavioral policy** $\pi_\beta$ if $\pi_\beta(a|\langle \boldsymbol{o}, b \rangle) = P^D(a|\boldsymbol{o}, b)$ for all actions $a$ and beliefs $b$. The next observation states that a policy matched by a DNN is executable if and only if the DNN is action sufficient. It requires a minor technical definition that rules out redundant parents: We say that a CBN is *action-minimal* if for every parent $X$ of action variable $A$ and every set $\boldsymbol{V}$ of variables disjoint from $A$ and $X$, we have $P(A|X, \boldsymbol{V}) \neq P(A|\boldsymbol{V})$. That is, there is no variable set $\boldsymbol{V}$ such that conditioning on $\boldsymbol{V}$ makes $X$ independent of its child $A$. Local minimality is entailed by the well-known stability/faithfulness conditions (Pearl, 2000, Ch.2.4).

**Observation 1.** *Let $D$ be a locally minimal dynamic decision network that matches a behavioral policy $\pi_\beta$. Then $\pi_\beta$ is executable if and only if the policy observation space $\boldsymbol{O}^{\pi_\beta}$ is action sufficient in $D$.*

Note that Observation 1 does not assume that a DDN matching $\pi_\beta$ is given. The import is that, as long as a behavioral policy is executable, it can be represented by an action sufficient DDN. Therefore for *executable behavioral policies, Theorem 1 applies, and what-if probabilities can be computed from conditional probabilities.*

**RL with executability.** As we explained in the introduction (Table 2), various common RL settings satisfy executability, so we may summarize our argument as follows:

On-policy learning OR online exploration OR complete observability $\Rightarrow$ executability
$\Rightarrow$ conditional policy values = what-if policy values = interventional probability policy values

where online exploration refers to any setting where the learning agent uses the behavioral policy to generate observations.

**RL without executability.** When none of the sufficient conditions hold, we have the partially observable offline off-policy (POOO) setting. *In the POOO setting, policy evaluation based on conditional probabilities*

*can, and typically does, lead to wrong policy values*, even in the asymptotic setting with perfect probability estimates. To highlight this important point, we discuss three examples in an informal way to convey the intuition for how confounders mislead policy estimation in the POOO setting.

*Scenario 1: Sports.* Consider the offline scenario of Figure 1b.

> Vancouver Canucks coach Rick Tocchet watches the Edmonton Oilers to learn from the season finalists. Based on a very large sample of games, he notices that whenever the Oilers shoot close to the goal, they score 50% of the time. So he directs the Canucks players to shoot whenever they get close. Tocchet is disappointed to find that the Canucks score only 25% of the time. "It must be that my players are worse than theirs", he thinks. Question: Is the coach right to blame his players?

The answer is no: Because Tocchet did not observe the health of the Oilers players, he did not realize that they shoot only when they are healthy. His policy directs the Canucks players to shoot whether they are healthy or not, which leads to a lower success rate. In this example, *using conditional probabilities leads to an overestimate of the Q-value of shooting.* The conditional probability is $1/2$ in the model (cf. Table 6). The correct Q-value estimate is given by the interventional probability of scoring given the intervention to shoot, which is $1/4$: Given that the intervention breaks the link between shooting and goalie health, the interventional probability is the joint probability that the player is healthy and that the goalie is not, which is $1/2 \times 1/2$.

*Scenario 2: Imitating Expert Drivers.* Consider a simplified version of the driving scenario of Zhang et al. (2020), where the expert driver brakes if and only if the car in front turns on its tail light (cf. Figure 5). In their scenario, the tail light is observable to the demonstrator but not to the imitator. Suppose that the real-world frequency of observing the tail light is 25%. The marginal behavioral policy then randomly brakes 25% of the time. Conditional probabilities misestimate the value of the marginal policy as follows.

- Since the expert hardly ever suffers an accident after not braking, the conditional Q-value of not braking is high. But the braking decisions of the imitator are independent of the tail lights of the car in front, so the probability of an accident after not braking is a substantial 25%, which is the interventional Q-value. Thus the conditional Q-value overestimates the interventional Q-value.

- When the imitator brakes, but the car in front does not, they slow down unnecessarily. Assuming the reward function includes the time to destination, the conditional Q-value is too low compared to the interventional Q-value.

*Scenario 3: The Frozen Lake.* Warrington et al. (2021) describe imitation learning in a frozen lake scenario where the expert has access to image data that show the locations of weak ice, but the trainee does not. In their simulations, the expert agent easily learns to cross the lake safely by taking into account where the ice is weak. The trainee observes that the expert never falls into the lake, and wrongly concludes that crossing a lake is a safe policy. In policy evaluation terms, they overestimate the value of crossing the lake compared to the policy of walking around it. In the simulations of Warrington et al. (2021), this policy mis-estimation leads to a catastrophically poor return for a naive imitation learning algorithm that ignores confounding.

# 6 Related Work: Current and New Research Directions

This section selectively describes current RL research, and some future directions, involving causal models with an emphasis on the distinction between observational, interventional, and counterfactual probabilities. We describe several directions for future research from the perspective of confounding and executability. For comprehensive surveys of causal RL, please see Bareinboim (2020), (Deng et al., 2023),Bareinboim et al. (2025) and (Schölkopf et al., 2021, Section E).

A common goal in previous research is leveraging a given causal model. Such approaches can be categorized as *causal model-based RL.* Causal model-based RL inherits the challenges and benefits of model-based RL

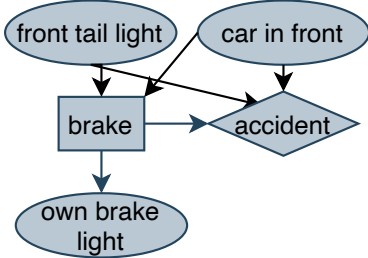

Figure 5: A car-driving scenario in which the state variable brake light is an effect of the braking action.

in general (Levine et al., 2020, Sec.5). Our discussion focuses on the following special features of causal model-based RL:

1. The ability to correctly evaluate the effects of interventions when confounders are present.

2. The greater expressive power of causal models, which define not only conditional reward-transition models, but also interventional and counterfactual reward-transition models.

3. The causal graph structure, which decomposes joint distributions into local mechanisms.

## 6.1 Online Causal RL

The term online RL in this section refers to single-agent online exploration, where the learning agent is the only agent executing a behavioral strategy to learn about the environment. Our analysis so far implies that in single-agent online learning, we can expect conditional probabilities to be unconfounded, which reduces the importance of advantage 1. However, the last two advantages (graphical structure and greater expressive power) apply to online learning as well, and have been leveraged in previous work on online RL.

**Causal Graphs and State Abstraction: Eliminating Irrelevant Variables**   Lemma 1 implies that conditioning on a superset $\boldsymbol{X} \supseteq pa_A$ of observable causes (parents) of $A$ suffices to ensure that conditional probabilities are causal. This means that the conditioning set need not include *all* observed variables, that is, it need not include the *entire* state observed by the agent. For example in the car driving model of Figure 5, for the braking decision, it suffices to condition on *FrontTaillight* and *CarinFront*; the *OwnBrakeLightVariable* can be ignored during the decision process. In general, a state variable $S$ is **conditionally irrelevant** to a decision if $S \perp R, \boldsymbol{S}'|A, Pa_A$. Thus a causal model supports variable selection as a form of state abstraction/simplification  (Peters et al., 2017, Sec.8.2.2). The paper by  Sen et al. (2017) is one of the first to leverage a given causal graph to reduce the effective state space. They prove that this reduction improves regret bounds in online bandit problems, when the bound is a function of the size of the state space. Zhang & Bareinboim (2020) provide an algorithm for reducing the state space by eliminating irrelevant variables given a causal model, which leads to substantive improvement in regret bounds. Wang et al. (2022; 2024) relate the elimination of irrelevant variables from a dynamic causal model to state abstraction.

**Causal Graphs and Imitation Learning**   The goal of behavioral cloning is to learn a policy that is similar to that of the behavior policy that generated the data. Using our notation, the goal is to learn a policy $\pi$ such that $b(A|\boldsymbol{S}) \approx \pi(A|\boldsymbol{S})$ where $b$ is the behavior policy to be imitated.  De Haan et al. (2019) in a paper on "causal confusion" point out that the conditional probability behavioral cloning objective can be problematic, even assuming executability. For example, in a self-driving car example like that of Figure 5, the observation signal includes the driver's own brake light. Since the brake light comes on only when the driver brakes, we have a very high correlation between braking and the brake light being on:

$$P(Brake = 1|BrakeLight = 1) \approx 1 \text{ and } P(Brake = 1|BrakeLight = 0) \approx 0.$$

Therefore if the imitator matches conditional probabilities according to this criterion, they will brake only if they observe the brake light coming on. In this scenario, the imitator will not respond to the location of the other cars and their tail lights, which fails to match the expert driver's behavior (and fails to avoid accidents). With knowledge of the causal graph Figure 5, we can eliminate the irrelevant brake light variable, as described in the previous paragraph, and avoid the causal confusion problem.

Zhang et al. (2020) introduce a different imitation objective, matching the return of a demonstrator. They introduce the concept of different observation spaces for the imitator and demonstrator, and state necessary and sufficient graphical conditions on a given causal graph for when an imitating policy can match the demonstrator's return, even when the observation signals of the imitator and demonstrator are different.

**Data Augmentation and Hindsight Counterfactuals** One of the traditional uses of models in RL, going back to the classic Dyna system (Sutton, 1990), is to augment the observed transition data with virtual experiences simulated from the model. Sun et al. (2024) utilize hindsight counterfactuals to generate virtual state transitions that specify the next state that would have occurred in the same scenario. These counterfactual state transitions take the form $P(\boldsymbol{S}'_{\hat{A}}|\boldsymbol{S}, \boldsymbol{S}', A)$, where we observe a next state transition from $\boldsymbol{S}$ to $\boldsymbol{S}'$ and ask what the next state would have been if the agent had selected action $\hat{A}$ instead of $A$.

Compared to traditional state-transition models of the form $P(\boldsymbol{S}'|\boldsymbol{S}, \hat{A})$, hindsight counterfactuals condition on more information and thus are potentially more accurate in generating virtual transitions. Sun et al. (2024) provide empirical evidence that hindsight state transitions speed up learning an optimal policy. Generating hindsight state transitions requires a causal model and is not possible with a traditional RL transition model that is based on conditional probabilities only.

**Learning A Causal Model from Online Data** Causal model discovery methods that are applicable to online RL learning include deep models based on auto-encoders (Lu et al., 2018) and GANs (Sun et al., 2024). Such deep generative models generate observations $\boldsymbol{x}$ from latent variables $\boldsymbol{z}$ (cf. Equation (2)), but they are not *structural* causal models based on a causal graph that represents local causal mechanisms. Huang et al. (2022) show how constraints from a given causal graph can be leveraged to learn latent state representations. Learning an influence diagram over state variables from online RL data seems to be a new research topic. An exciting new possibility for online learning is that the agent's exploration can include experimentation in order to ascertain the causal structure among the state variables.

## 6.2 Offline Causal RL

The off-policy evaluation (OPE) problem is to estimate the value function of a learned policy that is different from the behavioral policy. OPE is one of the major approaches to offline RL, where information about the behavioral policy is recorded in a previously collected dataset (Levine et al., 2020). The examples in this paper illustrated OPE based on a causal model (Section 4). In settings without executability, interventional policy evaluation is different from conditional policy evaluation, and research has therefore focused on such settings.

**Leveraging a Causal Model for OPE** Wang et al. (2021) investigate policy optimization based on interventional probabilities. To estimate the interventional probabilities when the behavioral policy is not executable, they assume that a causal model over the entire state space is available. The problem is then to compute a marginal interventional probability such as $P(SC|do(SH), CG)$ in our running example, from conditional probabilities over the entire state space, such as $P(SC|SH, CG, PH)$. Pearl's do-calculus provides powerful techniques for inferring marginal interventional probabilities from conditional probabilities, using what are known as *adjustment formulas*. Two well-known types of adjustment formula are the backdoor and the frontdoor criterion. Wang et al. (2021) utilize both to compute causal reward and state-transition probabilities from a given causal model, and show how to use the causal probabilities in an interventional Bellman equation. To illustrate the idea in our sports example, consider offline learning in the confounded model of Figure 1b. Since in this setting, only closeness-to-goal is observable, an executable policy would be based on this variable only (e.g. "shoot whenever you are close to the goal"). Finding interventional values for such a policy involves computing interventional probabilities such as $P(SC = 1|CG = 1, do(SH = 1))$.

According to the backdoor criterion, such probabilities can be computed by marginalizing over the unobserved values of player health as follows:

$$P(SC = 1|CG = 1, do(SH = 1)) =$$
$$P(PH = 1)P(SC = 1|PH = 1, CG = 1, SH = 1)) + P(PH = 0)P(SC = 1|PH = 0, CG = 1, SH = 1))$$
$$= 1/2 \cdot 1/2 + 1/2 \cdot 0 = 1/4$$

which agrees with the result of Table 6. While adjustment formulas provide an elegant approach to addressing spurious correlations in OPE, it is not entirely clear what their OPE use case is, as the learning agent does not have access to the latent variables that appear in the adjustment formulas. A possibility is that the behavioral agent uses their access to latent variables (e.g., the athlete has access to their health) to compute marginal interventional probabilities and communicate them to the offline learner.

**Causal OPE**   Most work is based on confounded MDPs (Zhang & Bareinboim, 2016; Kausik et al., 2024; Bruns-Smith, 2021): States are decomposed into an observed part and an unobserved part. The behavioral policy is assumed to depend on the complete state, whereas the policy to be evaluated depends on the observation signal only. A current line of research gives bounds on the extent of the bias due to spurious correlations, based on assumptions about the confounders (Kausik et al., 2024; Bruns-Smith, 2021). For example, "memoryless confounders" are sampled independently at each time instant. At the other extreme, time-invariant latent variables such as Goalie Health or Player Health in our example are known as "global confounders" (Kausik et al., 2024). Some of the causal OPE methods are model-based in the sense of estimating transition probabilities, but not in the sense of utilizing a dynamic SCM. Another difference is that the evaluation policies considered are Markovian in that they depend on current observations only (cf. Section 3.2). In contrast, our belief MDP framework allows for evaluation policies that depend on past observations/current beliefs.

Warrington et al. (2021) and Ghosh et al. (2021) propose effective Bayesian approaches to confounding, where the learning agent maintains a distribution over information that was used by the behavioral agent during training, but cannot be directly observed by the learning agent at test time (e.g., location of ice patches, target locations on a map). They do not use the confounded MDP framework. A valuable topic for future research is whether causal modeling would further improve the Bayesian approach.

**Learning a Causal Model from Offline Data**   One possibility is to learn a fully specified structural causal model with latent variables and use it for interventional policy evaluation (Lu et al., 2018). For learning a causal model offline, it is likely that the extensive work on learning Bayesian networks for temporal data can be leveraged, including recent approaches based on deep learning and continuous optimization (Sun et al., 2023). Latent variable models related to deep generative models are a promising direction for learning structural causal models (Geffner et al., 2022; Sun & Schulte, 2023; Mooij et al., 2016; Hyvärinen et al., 2010). Much of the research on learning a causal model assumes causal sufficiency (no confounders), but not all of it (Schölkopf et al., 2021). Learning an influence diagram with a causal graph over state variables from offline RL data seems to be a new research topic.

## 6.3   Offline + Online RL

In this hybrid setting, the learning agent interacts with the environment to collect more data, but a prior dataset is also utilized (Sutton, 1990; Janner et al., 2019). We can view this as an off-policy setting (see Figure 2a). A successful example is the AlphaGo system, which used an offline dataset of master games to find a good initial policy through imitation learning, then fine-tuned the policy with self-play (Silver et al., 2016). An active topic of research is how causal models can leverage offline datasets for online learning (Gasse et al., 2021; Zhang & Bareinboim, 2020), sometimes called generalized policy learning (Bareinboim, 2020).

Off-policy model-based RL approaches (Levine et al., 2020, Sec.5.2) can be applied with causal models to leverage the offline dataset. The techniques we outlined for causal models in the online and offline settings

can be utilized in the hybrid offline/online setting as well. For example, we can learn a causal graph from the offline dataset, and fine-tune it during online interactions with the environment. The ability to intervene in an environment in the online setting is potentially a powerful tool for learning a dynamic influence model. For instance, performing experiments can resolve the causal direction between two variables which may not be possible from observational data alone, at least not without assuming causal sufficiency.

Zhang & Bareinboim (2020) describe a hybrid approach for joint exploration and policy learning that is not based on a model. The offline dataset is used to estimate conditional state-reward transition probabilities. These estimates may not be correct interventional probabilities if confounders are present. Causal inference theory has established theoretical bounds on how far a conditional probability can differ from the interventional probability. Zhang and Barenboim use the resulting interval estimates for interventional probabilities as an input to optimistic exploration for online learning, where policies are evaluated according to their maximum possible value. Optimism under uncertainty is a well-known approach in RL for ensuring extensive exploration of the state space (Osband & Van Roy, 2014). The offline dataset provides tighter bounds on interventional probabilities than learning from online data only, thereby speeding up optimistic exploration.

## 7   Summary and Conclusion

We believe that many RL researchers share the intuition that in common RL settings, conditional probabilities correctly estimate the causal effects of actions. Our paper spelled out the conditions where we can expect conditional probabilities to correctly measure causal effects, defined by interventional probabilities. We provided a rigorous argument, using the formal semantics of causal models, for why these conditions lead to correct estimates. The key condition is *executability*: the observations available to the learning agent include the observations available to the behavioral agent, such that the learner can execute the behavioral policy. Our argument was that it is only in partially observable environments with offline off-policy learning (the POOO setting) that executability fails, and conditional probabilities can diverge from observed conditional probabilities. The reason is that in this learning setting, the environment may contain confounders, not observed by the learning agent, that influence both the decisions of the behavioral agent and the states and rewards that follow these decisions. Such confounders can introduce spurious correlations between decisions and states/rewards that do not correctly estimate the causal impact of the agent's decisions.

In contrast, if an RL setting satisfies executability, conditional probabilities coincide with interventional probabilities. Our argument for this conclusion involves two steps. (1) In such environments, the set of variables that the learning agent can observe is causally sufficient for the behavioral agent's actions, in the sense that it includes all common causes of the behavioral agent's decisions and other variables. (2) We prove formally, using causal theory, that if a set of observable variables is causally sufficient for actions, then actions are not confounded with states or rewards. Thus our analysis relies on the distinction between the observation signals available to the learning and behavioral agents, which has been highlighted by previous work in causal RL (Zhang et al., 2020; Zhang & Bareinboim, 2016; Kausik et al., 2024; Gasse et al., 2021).

In addition to the causal effects of interventions, causal models provide a rigorous specification of counterfactual probabilities through a formal semantics. Causality researchers have recently proposed using counterfactuals to enhance reinforcement learning (Bareinboim, 2020; Deng et al., 2023). We therefore extended our analysis to counterfactuals, distinguishing two kinds of counterfactuals: what-if queries (e.g. if I choose action $a'$ in state $s$ instead of action $a$, what is the likely reward?) and hindsight counterfactuals that condition on an observed outcome (e.g. given that I received reward $r$ after choosing action $a$ in state $s$, what is the likely reward if I choose action $a'$ instead?). We showed that under executability, what-if queries can be correctly estimated from conditional probabilities, but hindsight counterfactuals go beyond conditional probabilities (cf. Sun et al. (2024)).

Based on our analysis, we discussed the potential benefits of causal models in different reinforcement learning settings, such as online, offline, and off-policy. The most straightforward, though not the only, approach is to follow a model-based RL framework, replacing the traditional models involving conditional probabilities with a causal model. Structural causal models offer three main benefits: (1) They distinguish interventional and conditional probabilities, and therefore causation from correlation. (2) They factor the dynamics of a complex environment into local causal mechanisms represented in a causal graph. Local mechanisms are typically

invariant under interventions (Schölkopf et al., 2021), which means that a causal graph can help address the challenge of distribution shift (Levine et al., 2020), where a learned policy is evaluated against data gathered by another policy. (3) Causal models have greater expressive power than conditional probability models, since they can also evaluate interventional and counterfactual probabilities. We described existing work and promising future directions for how the benefits of causal models can be leveraged for reinforcement learning.

Reinforcement learning and causality are areas of AI and machine learning that naturally complement each other. The analysis in this paper provides a guide for reinforcement learning researchers as to when and how they can make use of causal concepts and techniques to advance reinforcement learning.

### Acknowledgements

We are indebted to the anomymous reviewers for very helpful suggestions. This research was supported by discovery grants to Poupart and Schulte from the Natural Sciences and Engineering Research Council of Canada.

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

## A  Online and Off-Policy Reinforcement Learning

In **online** reinforcement learning, the agent interacts with their environment through a behavioral policy to gather data. In the **on-policy** setting, the learned policy is evaluated using data generated by the learned policy itself. Figure 6 illustrates these learning settings with a diagram.

## B  Background: Markov Decision Processes

For temporal data the difference between causation and correlation stems from the possible presence of confounders—unobserved common causes of the agent's actions and other environment variables. The RL setting in which parts of the environment may be unobserved, which is known as a *partially observable Markov decision process* (POMDP). This section reviews the basic theory of POMDPs with respect to a new factored POMDP variant that facilitates applying causal concepts.

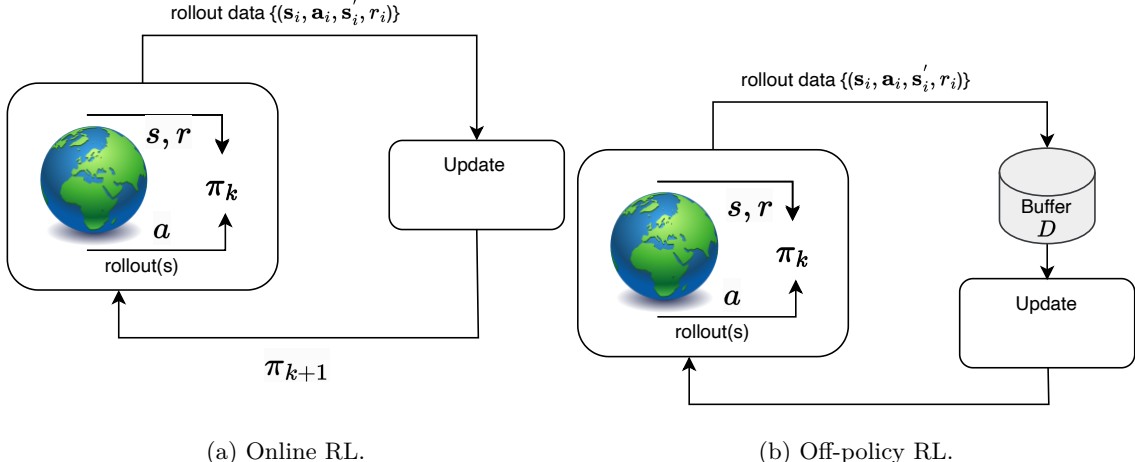

(a) Online RL.  (b) Off-policy RL.

Figure 6: Online RL settings; figures adapted from Levine et al. (2020). Figure 6a: In classic **online** RL, the policy $\pi_k$ is updated with streaming data collected by $\pi_k$ itself. Figure 6b: In classic **off-policy** RL, the agent's online experience is appended to a data buffer (also called a replay buffer) $D$, each new policy $\pi_k$ is used by the agent to collect additional data, such that $D$ comprises samples from $\pi_0, \ldots, \pi_k$, and all of this data is used to train an updated new policy $\pi_{k+1}$. *In both online settings the behavioral policy is executable,* because the policies used to generate the data are based on the same observations as the learned policy.

## B.1 Factored Partially Observable Markov Decision Processes

The state space of a **factored POMDP** is defined by a finite set of **state variables** $\boldsymbol{S}$. An environment **state** is an assignment $\boldsymbol{s}$ to the state variables. The state variables are partitioned as

$$\boldsymbol{S} = \boldsymbol{O} \cup \boldsymbol{Z}$$

where $\boldsymbol{O}$ represents the set of observable state variables, also called **the observation signal**, and $\boldsymbol{Z}$ represents the latent state variables, which we sometimes refer to simply as the latent state. If $\boldsymbol{Z}$ is empty, the state is **completely observable**; otherwise it is **partially observable**. Since a pair $(\boldsymbol{z}, \boldsymbol{o}) \equiv \boldsymbol{s}$ describes a state, we freely apply MDP notation to both $\boldsymbol{s}$ and pairs $(\boldsymbol{z}, \boldsymbol{o})$. In addition, we have an **action variable** $A$ ranging over a finite set of actions available to the agent, and a real-valued **reward variable** $R$.

A POMDP comprises the following components.

- An initial state distribution $P_E(\boldsymbol{s}_0)$, which can be factored into a distribution over latent and observed variables:
$$P_E(\boldsymbol{s}_0) \equiv P_E(\boldsymbol{z}_0, \boldsymbol{o}_0) = P_E(\boldsymbol{z}_0) \times P_E(\boldsymbol{o}_0|\boldsymbol{z}_0)$$
where $P_E(\boldsymbol{z}_0)$ is an *initial latent distribution* and $P_E(\boldsymbol{o}_0|\boldsymbol{z}_0)$ is the *initial observation model.*

- A *transition model*, which factors into a *latent state update model* and a *dynamic observation model* (see Figure 7a):

$$P_E(\boldsymbol{s}_{t+1}|\boldsymbol{s}_t, a_t) \equiv P_E(\boldsymbol{z}_{t+1}, \boldsymbol{o}_{t+1}|\boldsymbol{s}_t, a_t) = P_E(\boldsymbol{z}_{t+1}|\boldsymbol{s}_t, a_t) \times P_E(\boldsymbol{o}_{t+1}|\boldsymbol{z}_{t+1}, \boldsymbol{o}_t, a_t).$$

We assume that the current observations are Markovian in that they depend only on the current latent state, and the most recent observation and action:

$$P_E(\boldsymbol{o}_{t+1}|\boldsymbol{z}_{t+1}, \boldsymbol{o}_t, a_t) = P_E(\boldsymbol{o}_{t+1}|\boldsymbol{z}_{t+1}, \boldsymbol{o}_{\leqslant t}, a_{\leqslant t}) \tag{13}$$

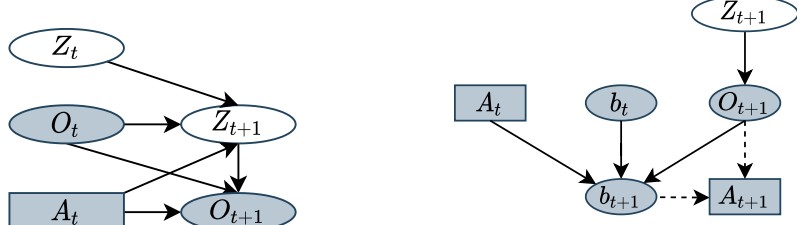

(a) Transition model in factored POMDP.

(b) Transition model and executable policy in belief MDP.

Figure 7: Generic causal graphs for POMDPs. Dashed lines represent a generic policy. Figure 7a: The environment transitions to a new latent state and generates an observation signal based on the current state and most recent action. Figure 7b: The agent selects an action through their policy based on their current belief state $b$. Their updated belief state depends on the previous belief state, previous action and new observation.

A **policy** $\pi$ (or agent function Russell & Norvig (2010)) maps a finite sequence of observations and actions to a distribution $P(A) \in \Delta(A)$ over actions:

$$\pi(a_t|o_0, a_0, o_1, a_1, \ldots, o_t) \equiv \pi(a_t|o_t, a_{<t}, o_{<t}).$$

A policy $\pi$ is **executable** if actions are independent of latent states, such that for all times $t$ we have

$$\pi(a_t|\boldsymbol{z}_t, \boldsymbol{o}_{\leqslant t}, \boldsymbol{a}_{<t}) = \pi(a_t|\boldsymbol{o}_{\leqslant t}, \boldsymbol{a}_{<t}). \tag{14}$$

## B.2 Posterior Updates

Given a current action $a_t$ and observation $\boldsymbol{o}_{t+1}$, the agent's beliefs move from current beliefs $b_t$ to updated beliefs $b_{t+1}$ through posterior updates; see Figure 7b. The observation signal provides an agent with information about the latent environment state through the posterior distribution $P(\boldsymbol{z}_{t+1}|\boldsymbol{o}_{\leqslant t+1}, a_{\leqslant t})$ where $\boldsymbol{z}_{t+1}$ is the current latent state, $\boldsymbol{o}_{\leqslant t+1}$ is the current observation sequence, and $a_{\leqslant t}$ is the previous observation sequence before taking the current decision. We derive the well-known POMDP formula for recursively updating the latent state posterior (Russell & Norvig, 2010, Ch.17.4.2) for a factored POMDP and an executable policy. For the purposes of the derivation, we define executability in terms of observation sequences as in Equation (14), rather than the epistemic state notion of Section 5, which provides a more general result.

**Observation 2.** *If the transition data are generated by a POMDP $P_E$ and an executable policy, the latent posterior update is given by*

$$P(\boldsymbol{z}_{t+1}|\boldsymbol{o}_{\leqslant t+1}, a_{\leqslant t}) = \alpha P_E(\boldsymbol{o}_{t+1}|\boldsymbol{z}_{t+1}, \boldsymbol{o}_t, a_t) \times \sum_{\boldsymbol{z}_t} P_E(\boldsymbol{z}_{t+1}|\boldsymbol{z}_t, \boldsymbol{o}_t, a_t) P(\boldsymbol{z}_t|\boldsymbol{o}_{\leqslant t}, a_{<t}) \tag{15}$$

*where $\alpha$ is a normalization constant.*

*Proof.*

$$P(\boldsymbol{z}_{t+1}|\boldsymbol{o}_{\leqslant t+1}, a_{\leqslant t}) \propto P_E(\boldsymbol{o}_{t+1}|\boldsymbol{z}_{t+1}, \boldsymbol{o}_{\leqslant t}, a_{\leqslant t}) \sum_{\boldsymbol{z}_t} P_E(\boldsymbol{z}_{t+1}|\boldsymbol{z}_t, \boldsymbol{o}_{\leqslant t}, a_{\leqslant t}) P(\boldsymbol{z}_t|\boldsymbol{o}_{\leqslant t}, a_{\leqslant t}) \text{ by Bayes' theorem}$$

$$= P_E(\boldsymbol{o}_{t+1}|\boldsymbol{z}_{t+1}, \boldsymbol{o}_t, a_t) \sum_{\boldsymbol{z}_t} P_E(\boldsymbol{z}_{t+1}|\boldsymbol{z}_t, \boldsymbol{o}_t, a_t) P(\boldsymbol{z}_t|\boldsymbol{o}_{\leqslant t}, a_{\leqslant t}) \text{ applying the Markov property Equation (13)}$$

$$= P_E(\boldsymbol{o}_{t+1}|\boldsymbol{z}_{t+1}, \boldsymbol{o}_t, a_t) \sum_{\boldsymbol{z}_t} P_E(\boldsymbol{z}_{t+1}|\boldsymbol{z}_t, \boldsymbol{o}_t, a_t) P(\boldsymbol{z}_t|\boldsymbol{o}_{\leqslant t}, a_{<t}) \text{ because } \pi \text{ is executable}$$

$\square$

**The Belief MDP** Even if the environment dynamics is Markovian in the state space, it may not be Markovian in observation space, because past observations can and typically do carry information about the current latent state. In order to apply MDP techniques to a POMDP, it is common to transform the POMD into an equivalent MDP whose states represent the agent's current beliefs. A **belief state** is a distribution $b(\boldsymbol{Z})$ over the latent environment state. The basic idea is to transform a POMDP into an MDP by replacing latent states with the agent's *beliefs* about latent states. As (Russell & Norvig, 2010, Ch.17.4.1.) write in their standard textbook: "The fundamental insight...is this: *the optimal action depends only on the agent's current belief state*" (emphasis Russell & Norvig).

**Belief Dynamics** For compactness, it is customary to assume that the agent's belief state at time $t$ conditions on the available observations and actions, so one writes $b_t(s) \equiv b_t(s|\boldsymbol{o}_{\leqslant t}, a_{<t})$. We adopt the standard POMDP notation for an agent's current belief state $b$ and $b'$ for a successor belief state. Similarly, we write $\boldsymbol{o}, \boldsymbol{o}'$ for an observation signal and its successor, and $\boldsymbol{z}, \boldsymbol{z}'$ for a latent state component and its successor. With these conventions, the posterior update (Equation (15)) becomes the belief update

$$b'(\boldsymbol{z}') = \alpha P_E(\boldsymbol{o}'|\boldsymbol{z}', \boldsymbol{o}, a) \times E_{\boldsymbol{z} \sim b(\boldsymbol{z})}[P_E(\boldsymbol{z}'|\boldsymbol{z}, \boldsymbol{o}, a)] \tag{16}$$

Equation (16) is analogous to the standard POMDP for atomic POMDPs, with the latent state variables $\boldsymbol{z}$ replacing the latent state $s$. A policy in a belief MDP maps a belief state and current observation to a distribution over actions (see Figure 7b and Section 4). The Bellman equation for a belief MDP is given by Equation (17).

## C Bellman Equations

The **Bellman equation for the observational Q-function** is as follows.

$$Q^{\pi,D}(\langle \boldsymbol{o}, b\rangle, a) = R^D(\langle \boldsymbol{o}, b\rangle, a) + \gamma \sum_{\boldsymbol{o}'} P^D(\boldsymbol{o}'|\langle \boldsymbol{o}, b\rangle, a)V^{\pi,D}(\boldsymbol{o}', b') \tag{17}$$

$$V^{\pi,D}(\langle \boldsymbol{o}, b\rangle) = \sum_{a} \pi(a|\langle \boldsymbol{o}, b\rangle)Q^{\pi,D}(\langle \boldsymbol{o}, b\rangle, a)$$

$$R^D(\langle \boldsymbol{o}, b\rangle, a) = E_{\boldsymbol{z} \sim b(\boldsymbol{z}|a)} \sum_{r} r \cdot P^D(R = r|\boldsymbol{z}, \boldsymbol{o}, a)$$

The $P^D$ terms are defined by a dynamic decision network as shown in Section 3. This is the same as the standard Bellman equation for belief MDPs (Russell & Norvig, 2010, Ch.17.4.3), with $\boldsymbol{z}$ in place of a latent $\boldsymbol{s}$ and the epistemic state $\langle \boldsymbol{o}, b\rangle$ in place of the belief state $b$.

The **Bellman equation for the interventional Q-function** is obtained from the observational Equation (17) by conditioning on the intervention $do(\hat{a})$ instead of the action $a$ as follows (Wang et al., 2021; Zhang & Bareinboim, 2020).

$$Q^{\pi,D}(\langle \boldsymbol{o}, b\rangle, do(\hat{a})) = R^D(\langle \boldsymbol{o}, b\rangle, do(\hat{a})) + \gamma \sum_{\boldsymbol{o}'} P^D(\boldsymbol{o}'|\langle \boldsymbol{o}, b\rangle, do(\hat{a}))V^{\pi,D}(\boldsymbol{o}', b') \tag{18}$$

$$V^{\pi,D}(\langle \boldsymbol{o}, b\rangle) = \sum_{\hat{a}} \pi(\hat{a}|\langle \boldsymbol{o}, b\rangle)Q^{\pi,D}(\langle \boldsymbol{o}, b\rangle, do(\hat{a}))$$

$$R^D(\langle \boldsymbol{o}, b\rangle, do(\hat{a})) = E_{\boldsymbol{z} \sim b(\boldsymbol{z}|do(\hat{a}))} \sum_{r} r \cdot P^D(R = r|\boldsymbol{z}, \boldsymbol{o}, do(\hat{a}))$$

The what-if Bellman Equation (10) is obtained from the interventional Equation (18) by adding to the original observation signal $\langle \boldsymbol{o}, b \rangle$ the observed act $a$, resulting in the expanded observation signal $\langle \boldsymbol{o}, a, b \rangle$.

## D  Proofs

### D.1  Proof of Theorem 1

**Theorem 1.** *Suppose that an observation signal $\boldsymbol{O}$ is action sufficient in a dynamic probabilistic SCM D. Then for every epistemic state, observed act $a$, and action $\hat{a}$:*

$$Q^{\pi,D}(\langle \boldsymbol{o}, a, b \rangle, do(\hat{a})) = Q^{\pi,D}(\langle \boldsymbol{o}, b \rangle, do(\hat{a})) = Q^{\pi,D}(\langle \boldsymbol{o}, b \rangle, \hat{a}).$$

*That is, given that the direct causes of an act have been observed, the Q-value associated with an intervention $\hat{a}$ replacing the actual act $a$ equals the Q-value associated with conditioning on $\hat{a}$, independent of the act.*

We prove the result from the basic definitions rather than using derived principles like Pearl's do-calculus. The proof involves two main steps.

1. Show that in any causal model, if the parents of an intervened variable $A$ are observed, then interventional, causal, and what-if probabilities are equivalent. Intuitively, if the parents of $A$ are conditioned on, then removing the links between $A$ and its parents does remove any noncausal correlations.

2. Therefore under action sufficiency, causal and conditional reward and transition probabilities are equivalent. Since reward and transition probabilities determine Q-values by the Bellman equation, the equivalence of causal and conditional Q-values follows.

**Lemma 1.** *Let $C$ be a probabilistic SCM and let $\boldsymbol{Y}, A, \boldsymbol{X}$ be a disjoint set of random variables such that $\boldsymbol{X}$ includes all parents of $A$ except for possibly a noise variable $U_A$ of $A$, and none of the descendants of $A$. Then for any actions $a, \hat{a}$ we have*

$$P^C(\boldsymbol{Y}|\boldsymbol{X} = \boldsymbol{x}, A = a, do(A = \hat{a})) = P^C(\boldsymbol{Y}|\boldsymbol{X} = \boldsymbol{x}, do(A = \hat{a})) = P^C(\boldsymbol{Y}|\boldsymbol{X} = \boldsymbol{x}, A = \hat{a}).$$

*Proof.* The source variable posterior satisfies the following independence conditions:

$$b(\boldsymbol{u}, u_A|\boldsymbol{X} = \boldsymbol{x}, A = a) = b(\boldsymbol{u}|\boldsymbol{X} = \boldsymbol{x}, A = a) \cdot b(u_A|\boldsymbol{X} = \boldsymbol{x}, A = a)$$
$$b(\boldsymbol{u}|\boldsymbol{X} = \boldsymbol{x}, A = a) = b(\boldsymbol{u}|\boldsymbol{X} = \boldsymbol{x})$$
$$b(\boldsymbol{u}, u_A|\boldsymbol{X} = \boldsymbol{x}) = b(\boldsymbol{u}|\boldsymbol{X} = \boldsymbol{x}) \cdot b(u_A|\boldsymbol{X} = \boldsymbol{x})$$

The first independence holds because $\boldsymbol{X}$ contains all the parents of $A$ (i.e., $\boldsymbol{X}, A = a$ contains the entire Markov blanket of $u_A$). The second independence holds because $\boldsymbol{X}$ contains all the parents of $A$ and none of its descendants, so by the Markov condition, $A$ is independent of all its non-descendants. Since $\boldsymbol{u}$ contains only source variables, it contains no descendent of $A$. Similarly, the third independence holds because $u_A$ is independent of all its non-descendants, and $\boldsymbol{X}$ contains no descendent of $A$ and hence no descendant of $u_A$.

Now consider the evaluation of the counterfactuals $P^C(\boldsymbol{Y}|\boldsymbol{X} = \boldsymbol{x}, A = a, do(A = \hat{a}))$ and $P^C(\boldsymbol{Y}|\boldsymbol{X} = \boldsymbol{x}, do(A = \hat{a}))$. Each probability is calculated in the same submodel $\mathcal{S}_{\hat{a}}$ but with different posteriors. Let $C_1$ be the submodel with source posterior distribution $b(\boldsymbol{u}, u_A|\boldsymbol{X} = \boldsymbol{x}, A = a)$ and let $C_2$ be the submodel with source posterior distribution $b(\boldsymbol{u}, u_A|\boldsymbol{X} = \boldsymbol{x})$. In each submodel, $A$ is not generated by source variables but manipulated to the value $\hat{a}$. Let $\boldsymbol{U}$ be the set of source variables other than $U_A$. For an assignment of values to variables $\boldsymbol{W} = \boldsymbol{w}$, where $A \notin \boldsymbol{W}$, let $U_{\boldsymbol{w}|a}$ be the set of assignments to the source variables $\boldsymbol{U}$ such that the recursive solution procedure generates the assignment $\boldsymbol{W} = \boldsymbol{w}$ if variable $A = a$. Together with the independence conditions above, we therefore have the following:

$$P^{C_1}(\boldsymbol{Y} = \boldsymbol{y}, \boldsymbol{X} = \boldsymbol{x})$$

$$= \sum_{\boldsymbol{u} \in U_{\boldsymbol{x}, \boldsymbol{y}|\hat{a}}} \sum_{u_a} b(\boldsymbol{u}, u_A | \boldsymbol{X} = \boldsymbol{x}, A = a)$$

$$= \sum_{\boldsymbol{u} \in U_{\boldsymbol{x}, \boldsymbol{y}|\hat{a}}} b(\boldsymbol{u}|\boldsymbol{X} = \boldsymbol{x}) \sum_{u_a} b(u_A | \boldsymbol{X} = \boldsymbol{x}, A = a)$$

$$= \sum_{\boldsymbol{u} \in U_{\boldsymbol{x}, \boldsymbol{y}|\hat{a}}} b(\boldsymbol{u}|\boldsymbol{X} = \boldsymbol{x}) \cdot 1$$

$$= \sum_{\boldsymbol{u} \in U_{\boldsymbol{x}, \boldsymbol{y}|\hat{a}}} b(\boldsymbol{u}|\boldsymbol{X} = \boldsymbol{x}) \sum_{u_a} b(u_A | \boldsymbol{X} = \boldsymbol{x})$$

$$= P^{C_2}(\boldsymbol{Y} = \boldsymbol{y}, \boldsymbol{X} = \boldsymbol{x})$$

Since the joint probabilities are the same for each posterior, so are the conditional counterfactual probabilities, which establishes the first equality of the Lemma.

The argument for the second equality is as follows: Given the assignment $pa^{\boldsymbol{x}}$ induced by the observations $\boldsymbol{x}$, we can define a conditional probability over actions by $P(a|\boldsymbol{x}) = P(a|pa^{\boldsymbol{x}}) = \sum_{u_A : f_A(pa^{\boldsymbol{x}}, u) = a} b(u_A)$, that is, summing over the set of noise variable variables that generate the observed action. The conditional and interventional distributions differ only by this term, which does not depend on the target $\boldsymbol{Y}$ and therefore cancels out in the conditional probability. $\square$

Our next proposition asserts the equivalence for the reward and transition models under executability.

**Proposition 1.** *Suppose that an observation signal $\boldsymbol{O}$ is action sufficient in a dynamic decision network $D$. Then for every epistemic state, observed act $a$, and action $\hat{a}$:*

$P^D(R|\boldsymbol{O}, A, do(\hat{A})) = P^D(R|\boldsymbol{O}, do(\hat{A})) = P^D(R|\boldsymbol{O}, \hat{A})$ *and* $P^D(\boldsymbol{S}'|\boldsymbol{O}, A, do(\hat{A})) = P^D(\boldsymbol{S}'|\boldsymbol{O}, do(\hat{A})) = P^D(\boldsymbol{S}'|\boldsymbol{O}, \hat{A})$. *That is, the conditional, what-if interventional reward/transition probabilities are the same.*

*Proof.* Follows immediately from Lemma 1. $\square$

Theorem 1 follows from Proposition 1.

### D.2 Proof of Observation 1

**Observation 1.** *Let $D$ be a locally minimal dynamic decision network that matches a behavioral policy $\pi_\beta$. Then $\pi_\beta$ is executable if and only if the policy observation space $\boldsymbol{O}^{\pi_\beta}$ is action sufficient in $D$.*

*Proof.* ($\Rightarrow$): Suppose that every parent of $A$ is observed (i.e., $Pa_A \subseteq \boldsymbol{O}$). By Definition 1(3), the only potential descendant of $A$, except for successor variables, is the reward variable $R$. Thus the set of observed variables $\boldsymbol{O}$ contains no descendant of $A$. By the Markov condition, $A$ is independent of all non-descendants given the parents of $A$. So given $\boldsymbol{O}$, $A$ is independent of all contemporaneous latent environment variables $\boldsymbol{Z}$, which is the definition of an executable policy. The same argument applies to the successor action variable $A'$.

($\Leftarrow$): Suppose that $A$ is independent of the latent variables $\boldsymbol{Z}$ given the observed variables $\boldsymbol{O}$. Then action-minimality requires that no latent variable is a parent of $A$, which is the definition of action sufficiency. $\square$

## E    Hindsight Counterfactuals and Policy Evaluation.

A remarkable feature of counterfactual hindsight probabilities is that they can differ from conditional probabilities even in action sufficient settings, such as online learning. This subsection gives examples to illustrate

the phenomenon, two examples for the reward model and one for policy evaluation. The general insight is that while it has long been noted in RL that *past* observations allow us to infer current latent states (Kaelbling et al., 1998; Hausknecht & Stone, 2015), *future* information also allows us to infer current latent states.

Suppose that in observed state $\boldsymbol{o}$, an act $a$ was followed by a reward $r$. We can then ask "what would the reward have been if the agent had chosen the action $do(\hat{a})$" instead? The corresponding **hindsight reward probability** is given by counterfactual queries of the form $P(R_{do(\hat{a})}|\boldsymbol{O}, A, R)$.

A simple reward hindsight query in our sports example would be

$$P^D(SC_{SH=1} = 1 | CG = 1, PH = 1, SH = 1, SC = 1),$$

where $SC_{SH=1} = 1$ denotes the potential outcome variable "scoring a goal after intervening to shoot".

To evaluate the hindsight reward probability in the online model of Section 3.2, first we update the initial belief given the observations:

$$b(GH = 0 | CG = 1, PH = 1, SC = 1) = 1.$$

Informally, since the player scores only if the goalie is not healthy, we can infer from their scoring that the goalie is not healthy. Given that the player is healthy and the goalie is not, the player is certain to score, so the hindsight reward probability is 1:

$$P^D(SC_{SH=1} = 1 | CG = 1, PH = 1, SC = 1) = P(SC = 1 | CG = 1, PH = 1, GH = 0, do(SH = 1)) = 1$$

Without hindsight, the chance of scoring is only $1/2$, since the goalie has a 50% chance of being healthy.

A more interesting example is to consider not only immediate rewards, but hindsight based on future rewards (as in hindsight credit assignment (Harutyunyan et al., 2019)). Suppose that a player does not take a shot, and then their team scores at the next time instant. Since this implies that the goalie is not healthy, we can infer that they would have scored if they had taken a shot earlier. Using counterfactual notation, we have

$$P^D(SC_{SH=1} = 1 | CG = 1, PH = 1, SH = 0, SC' = 1) = P(SC = 1 | CG = 1, PH = 1, GH = 0, do(SH = 1)) = 1,$$

where as before $SC_{SH=1} = 1$ denotes the potential outcome variable "scoring a goal after intervening to shoot", and $SC' = 1$ denotes that the observed outcome at the *next* time step is that the player scored.

Hindsight Q-values can be defined by including the observed outcomes as part of the agent's observation signal. Figure 8 shows how Q-values change with hindsight. If the player is made not to shoot, they have a 50% chance of maintaining possession. If they maintain possession, they will shoot at the next step according to the behavioral policy, so they are certain to score then because the goalie is not healthy. Hence their expected return after not shooting is $1/2$; in symbols

$$Q^D(\langle CG = 1, PH = 1, SC = 1, b \rangle, do(SH = 0)) = 1/2.$$

These examples illustrate how observing outcomes can be a powerful source of information about the latent environment state (goalie health in our example).

## F   Computation of Reward Probabilities

We use the causal graphs of Figure 1 parametrized as in Table 5. The computations are shown in Figures 9 to 11.

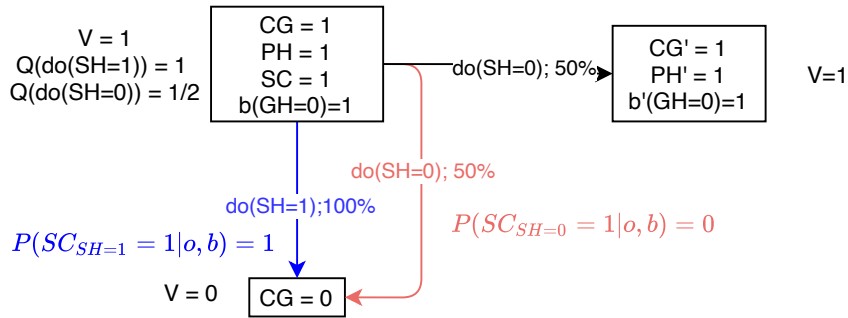

Figure 8: On-policy policy evaluation in the online setting with hindsight. The policy evaluated is the standard behavioral policy (shoot if and only if close to the goal and healthy). The evaluation uses interventional reward and transition probabilities derived from the online DDN of of Section 3.2. The diagram shows the V value and Q action values for the epistemic state where the agent is observed to be close to the goal, their belief is uniform over the latent variables, and *the observation signal includes the current reward (goal scored)*.

## G   Computation of Policy Values

Figure 12 is a state diagram that gives the Q and V values for the marginal policy Equation (12) based on (i) conditional probabilities (Figure 12a) and ii) based on interventional probabilities (Figure 12b). Due to non-executability, *they are different for the epistemic state where $CG = 1$*. For the conditional action values, we have $Q^D(CG = 1, b, SH = 1) = 1/2$. The epistemic state value $V$ satisfies the equation $V = 1/4 + 1/4V$, so $V = 1/3$.

For the interventional action values we have $Q^D(CG = 1, b, do(SH = 1)) = 1/4$. The epistemic state value $V$ satisfies the equation $V = 1/2 \cdot 1/4 + 1/4V$, so $V = 1/6$.

Figure 13 shows the computation of values for the marginal policy $\mu$, given that we observe a player not taking a shot when they are close to the goal. The confounded model of Figure 1b implies that the player is not healthy. Since they score only if they are healthy, it follows that their expected reward is 0 regardless of what action we direct them to perform.

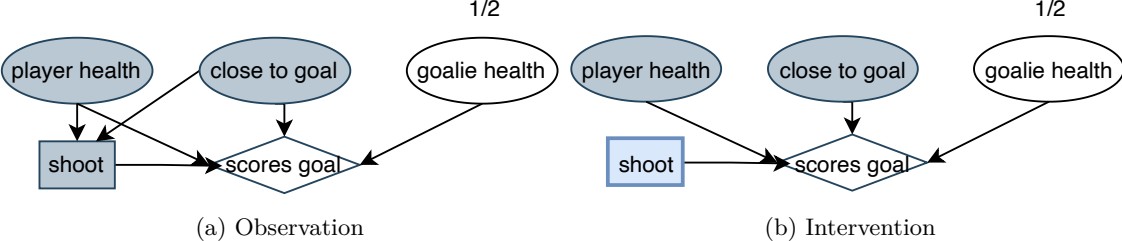

Figure 9: Observational and intervention probabilities in the online model of Figure 1a. Gray indicates observed variables whose values are specified in the query. Numbers indicate posteriors over latent variables, given the observations. Light blue indicates an intervention on a variable. Figure 9a: The observational scoring probability $P(SC = 1|CG = 1, SH = 1, PH = 1)$ is 1/2, the same as the probability that the goalie is healthy. Figure 9b: The query $P(SC = 1|CG = 1, PH = 1, do(SH = 1))$ is evaluated in the intervention model. The scoring probability remains 1/2, because both player health and shooting are observed, so breaking the causal link between them has no effect on the scoring probability.

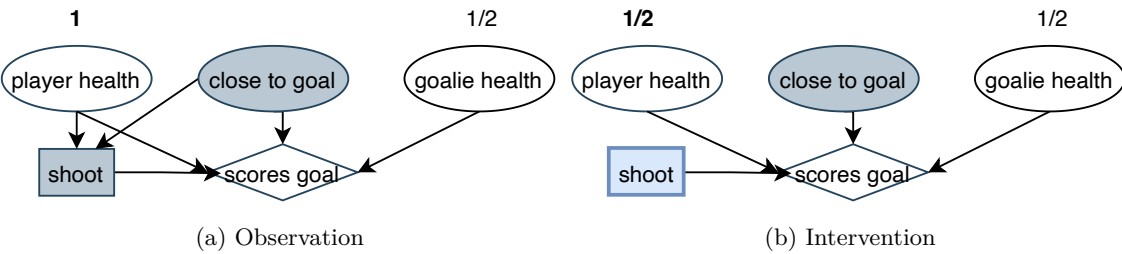

Figure 10: Observational and intervention probabilities in the confounded offline model of Figure 1b. The query $P(SC = 1|CG = 1, SH = 1)$ is evaluated in the observation model Figure 10a. *If we see a player shooting, we can infer that they are healthy.* Therefore the player scores if and only if the goalie is not healthy, so the scoring probability is $P(SC = 1|CG = 1, SH = 1) = 1/2$. The query $P(SC = 1|CG = 1, do(SH = 1))$ is evaluated in the intervention model Figure 10b. Without a link between player health and shooting, the probability of player and goalie health are both 1/2, which means that the scoring probability is 1/4.

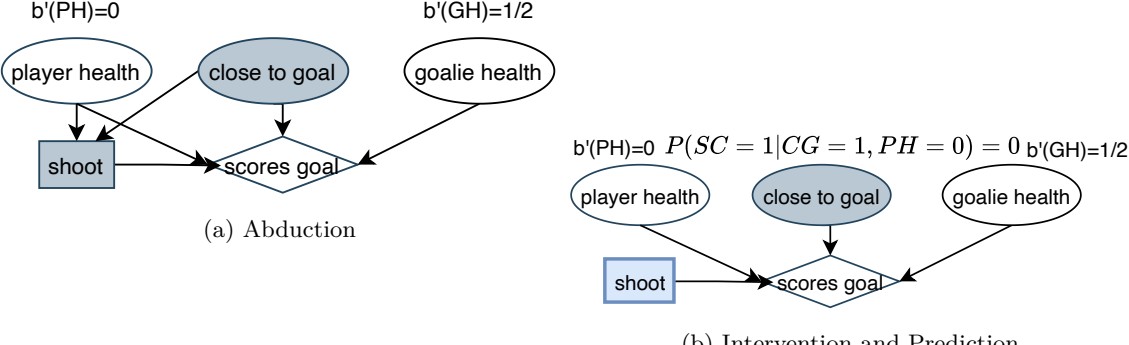

Figure 11: Evaluating the what-if counterfactual query $P(SC = 1|CG = 1, SH = 0, do(SH) = 1))$ for the confounded offline model of Figure 1b. Numbers indicate posterior probabilities of latent source variables given the query observations. Figure 11a, Abduction: The posterior probability of the player being healthy is 0, given that they did not shoot. Figure 11b, Intervention and Prediction: The truncated model removes the link between Player Health and shooting and uses the posterior distribution over source variables. In the truncated model, the scoring probability is 0, given that we have inferred that the player is not healthy.

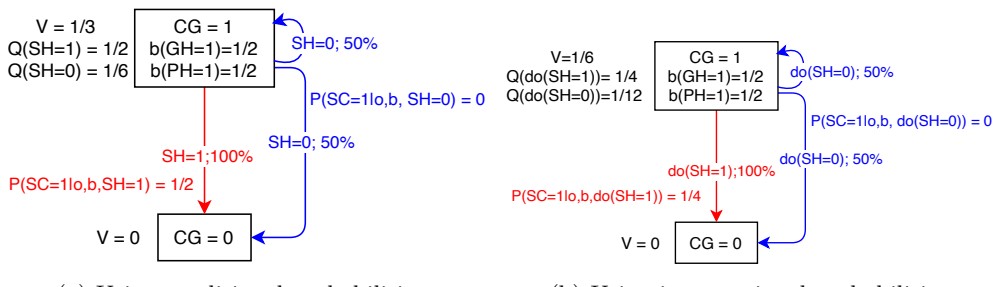

(a) Using conditional probabilities.  (b) Using interventional probabilities.

Figure 12: Off-policy policy evaluation in the offline setting of Figure 1b. An epistemic state comprises values for the observable variable *CG* and a belief over the values of the latent variables *GH* and *PH*. $CG = 0$ is an absorbing state (once the team loses possession, the attack is over; see Section 3.2). The policy evaluated is the marginal behavioral policy of Equation (12), which chooses to shoot with probability $1/2$ if the player is close to the goal. The evaluation uses conditional resp. interventional reward and transition probabilities derived from the DDN of Section 3.2. The diagram shows the V value and Q action values for the epistemic state where the agent is observed to be close to the goal, and their belief is uniform over the latent variables. Transitions are labelled with probabilities. State-action pairs are annotated with expected rewards. Figure 12a: Policy evaluation using *conditional probabilities* as in Equation (17). Figure 12b: Policy evaluation using *interventional probabilities* as in Equation (18).

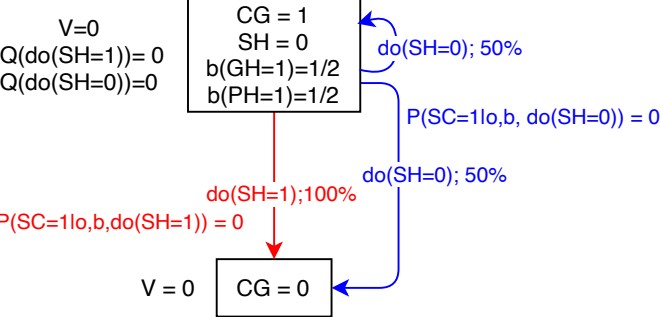

Figure 13: Off-policy policy evaluation based on what-if counterfactuals in the confounded offline model of Figure 1b. The policy evaluated is the marginal behavioral policy. The diagram shows the V value and Q action values for the epistemic state where the agent is observed to be close to the goal, their belief is uniform over the latent variables, and *they do not take a shot*.

