# OpenReview forum: "When Should Reinforcement Learning Use Causal Reasoning?"
_TMLR — Accepted by TMLR_

### Review · Reviewer_6nnX · 2024-10-03

**Summary Of Contributions:**

The paper did a comprehensive discussion on searching the scenario such that conditional causal modeling is useful to reinforcement learning.

**Audience:**

Yes

**Broader Impact Concerns:**

The paper didn't provide Broader Impact Statement.

**Claims And Evidence:**

Yes

**Requested Changes:**

(1) Please think about how to optimize space usage. Current locations of Figures and the related text are difficult to read.
(2) Please add more explanation after  equations .

**Strengths And Weaknesses:**

Pro. (1) The discussion looks comprehensive.

Con. (1) Conclusions is not clear and the paper didn't flow well, causing readability issue.

---

> ### Author Response · Authors · 2025-01-04
> **Revision and Replies**
>
> Thank you for your comments, which were great feedback for improving our submission. Please see the new revision and its change log. We appreciate the time you put into reading our submission. We begin with addressing the high-level concerns, then move to line-by-line replies.
>
> HIGH-LEVEL CONCERNS
>
> Paper length. The main reasons for the length of the paper are as follows:
>
> •	comprehensiveness (treating a substantive number of RL settings as well as causal concepts), noted as a strength by reviewer 2
> •	making the paper self-contained for readers from different backgrounds (RL researchers and causality theorists) and who seek different levels of detail
> •	using figures and worked examples to illustrate concepts for unfamiliar readers.
>
> As a result of these design choices, we believe that our paper is long but not dense.
>
> Using the reviewer suggestions, we have reduced the length by three pages, from 38 pages to 35 pages (details below), while adding some new material to address reviewer concerns. Even more importantly, we have simplified the concepts and increased the modularity, meaning that it is now even easier for readers to focus on the parts that are relevant to them. We describe these changes in some detail below.
>
> We also would like to make a suggestion: the analysis of interventions and counterfactuals is a natural partition point for our discussion. How about dividing the paper into two parts? The first part could be called “When Should Reinforcement Learning Use Causal Reasoning? Part I: Interventions” and would run at about 22 pages (excluding references and appendices). The second part could be called “When Should Reinforcement Learning Use Causal Reasoning? Part II: Counterfactuals” and would run at about 17 pages (excluding references and appendices). The parts could be (re)-reviewed separately by different reviewers to reduce the reviewing burden.
>
> In any case, we have rewritten the paper in several ways to increase modularity. Added text is shown in the red in the revised version.
>
> 1.	Rewriting the introduction. The main message of our paper is that causal reasoning is needed if and only if the behavioral policy, which generates training data, utilizes information that is not available to the learning agent, who aims to discover an optimal policy. We have rewritten the introduction so that it summarizes the main message at a high-level, and gives the intuition for what our argument for the message is. We also moved to the introduction our discussion, with examples and references, of which RL settings exhibit the information asymmetry between the behavioral and the learned policy. (This includes the train-test asymmetry stemming from generalizing from limited data mentioned by reviewer 1, thank you for this great reference.) In sum, we believe that the introduction is sufficient for readers who just want to get the main message and the relevant intuitions.
>
> 2.	Adding section summaries. We have added a figure (fig 4) that summarizes the main result/concept of each section and shows the dependencies between each section. Also the section’s main result is summarized at the beginning and end of each section. Thus if the details of a section are not relevant to a reader, they can take in the summaries and continue.
>
> We have rewritten the paper to simplify the conceptual framework as follows.
>
> 1.	We have merged the concepts of observation-equivalence and an executable (behavioral) policy. Now all results are stated in terms of executability only.
> 2.	Instead of first introducing general POMPD policies and then policies based on belief states, we immediately move to belief MDPs.
> 3.	We have eliminated the discussion of how structural causal models compared to deep generative models.
>
> LINE-BY-LINE REPLIES.
>
> Added text is shown in the red in the revised version.
>
> **“Con. (1) Conclusions is not clear and the paper didn't flow well, causing readability issue.”**
>
> We expanded the introduction with a high-level summary of our main conclusion and with more detailed discussion of the consequences for RL settings. We added the conclusion/main result of each section at the beginning and end of each section.
>
> **“Please add more explanation after equations .”**
>
> We have added explanations of notation after equations. Notation is now explained either right above or right below the equation. (An exception are the various Bellman equation variants, we did not repeat explanations of their notation). For theorems, brief explanations of the equation’s import. Please note also that Table 2 contains a summary of the notation used throughout the paper.
>
> **“Current locations of Figures and the related text are difficult to read.”**
>
> We have reviewed and changed figure locations. However, because Latex automates float placement, optimizing it is best done by the final typesetter.

---

### Review · Reviewer_UboP · 2024-11-19

**Summary Of Contributions:**

The paper presents a thorough analysis of causal models in different reinforcement learning regimes. The authors analyze under what assumptions causally factored methods will differ from those that do not take into account the causal structure of the environment.
The authors conclude that specifically in the partial observable offline regime it is important to take into account additional causal structure.

**Audience:**

Yes

**Claims And Evidence:**

Yes

**Requested Changes:**

Please discuss the assumptions on the available samples for the agent to estimate probabilities.

Please tighten the writing overall.

**Strengths And Weaknesses:**

### Strengths

The paper is very thorough and presents an interesting analysis of different scenarios in which causal and regular probability estimates differ or don't in RL.
It serves both as a good introduction and survey of the notion of causal graphs and do-calculus in the context of RL, and presents interesting clarifying insights in its own right.

I checked the math as far as possible for me, however I am not too familiar with causal inference and some detailed errors might have slipped past me. As all results seem relatively intuitive to me though, I am not too concerned that there might be major errors.

### Weaknesses

An implicit assumption that is made throughout the paper, and as far as I can tell remains unstated, is that all analysis are done in an infinite sample regime in which all probabilities can be estimated correctly. In cases where online exploration and limited samples are a factor, I do not believe the authors conclusions hold. E.g. if a state has only ever been seen once, the agent cannot be sure what to credit any received reward for: its actions or idiosyncrasies of the state.

Recent work [1] has argued that under exploration, any MDP turns into a POMDP, which means that most such problems (except in the unlimited sample regime) might benefit from causal structures.

I think discussing this assumption and the implications of limited exploration/samples on the recommendations made could strengthen the value of the paper for RL researchers. Many might be more used to considering sample efficiency in addition to inference modelling. Note that I do not claim that the analysis in the paper is _wrong_, just that a discussion on this would greatly improve its usefulness to the community. Overall, analyzing the expressiveness of different models without accounting for sample complexity is still a very valuable contribution in its own right.

An additional question I have is whether discovering the causal structure of the environment is feasible in the strict POOO regime? If not, then this strictly necessitates pre-specifying the causal graph? If yes, then this is an important limitation for this setting in terms of learnability, that should be highlighted for the community.

Minor comment: The paper is extremely long. The core results could be presented in a shorter fashion. Several tables and figures, as well as examples take up much room and could potentially be cut. For example, pages 3 and 4 take up a lot of space to present very simple well-known concepts to the reader.

As another example, the "related work" section takes up 5 and a half pages, which is enormous for a paper that does not aim to be a survey paper. Similarly, 7 1/2 pages of background is very thorough for a non-survey paper.

The length of the paper does detract from the quality somewhat, as I found myself getting somewhat lost while reading it. With a shorter paper that reaches its conclusion quicker, its easier for readers to follow.

[1] Why Generalization in RL is Difficult: Epistemic POMDPs and Implicit Partial Observability, Gosh et al., NeurIPS 21

---

> ### Author Response · Authors · 2025-01-20
> **Revision and High-level Replies 1/2**
>
> Thank you for your comments, which were great feedback for improving our submission. We appreciate the time you put into reading our submission. We begin with addressing the high-level concerns, then move to line-by-line replies in our second response.
>
> HIGH-LEVEL CONCERNS
>
> Paper length. The main reasons for the length of the paper are as follows:
>
> •	comprehensiveness (treating a substantive number of RL settings as well as causal concepts), noted as a strength by reviewer 2
> •	making the paper self-contained for readers from different backgrounds (RL researchers and causality theorists) and who seek different levels of detail
> •	using figures and worked examples to illustrate concepts for unfamiliar readers.
>
> As a result of these design choices, we believe that our paper is long but not dense.
>
> Using the reviewer suggestions, we have reduced the length by three pages, from 38 pages to 35 pages (details below), while adding some new material to address reviewer concerns. Even more importantly, we have simplified the concepts and increased the modularity, meaning that it is now even easier for readers to focus on the parts that are relevant to them. We describe these changes in some detail below.
>
> We also would like to make a suggestion: the analysis of interventions and counterfactuals is a natural partition point for our discussion. *How about dividing the paper into two parts?* The first part could be called “When Should Reinforcement Learning Use Causal Reasoning? Part I: Interventions” and would run at about 22 pages (excluding references and appendices). The second part could be called “When Should Reinforcement Learning Use Causal Reasoning? Part II: Counterfactuals” and would run at about 17 pages (excluding references and appendices). The parts could be (re)-reviewed separately by different reviewers to reduce the reviewing burden.
>
> In any case, we have rewritten the paper in several ways to increase modularity. Added text is shown in the red in the revised version.
>
> 1.	Rewriting the introduction. The main message of our paper is that causal reasoning is needed if and only if the behavioral policy, which generates training data, utilizes information that is not available to the learning agent, who aims to discover an optimal policy. We have rewritten the introduction so that it summarizes the main message at a high-level, and gives the intuition for what our argument for the message is. We also moved to the introduction our discussion, with examples and references, of which RL settings exhibit the information asymmetry between the behavioral and the learned policy. (This includes the train-test asymmetry stemming from generalizing from limited data mentioned by reviewer 1, thank you for this great reference.) In sum, we believe that the introduction is sufficient for readers who just want to get the main message and the relevant intuitions.
> 2.	Adding section summaries. We have a figure (fig 4) that summarizes the main result/concept of each section and shows the dependencies between each section. Also the section’s main result is summarized at the beginning and end of each section. Thus if the details of a section are not relevant to a reader, they can take in the summaries and continue.
>
> We have rewritten the paper to simplify the conceptual framework as follows.
>
> 1.	We have merged the concepts of observation-equivalence and an executable (behavioral) policy. Now all results are stated in terms of executability only.
> 2.	Instead of first introducing general POMPD policies and then policies based on belief states, we immediately move to belief MDPs.
> 3.	We have eliminated the discussion of how structural causal models compared to deep generative models.

---

> > ### Author Response · Authors · 2025-01-20
> > **Revision and High-level Replies 2/2**
> >
> > LINE-BY-LINE REPLIES.
> >
> > *Added text is shown in the red in the revised version.*
> >
> > **“An implicit assumption that is made throughout the paper, and as far as I can tell remains unstated, is that all analysis are done in an infinite sample regime in which all probabilities can be estimated correctly.”**
> >
> > Correct. We now state this assumption in the introduction (page 4).
> >
> > **“Recent work [1] has argued that under exploration, any MDP turns into a POMDP, which means that most such problems(except in the unlimited sample regime) might benefit from causal structures.”**
> >
> > This is a great point which we were happy to add in the introduction where we discuss RL settings that might benefit from causal structures. The additional information available during training, but only partially during test time  can and typically does act as a confounder. We have expanded in the introduction the discussion of RL settings where the information asymmetry between training (behavioral policy) and testing (learned policy) leads to confounding problems.
> >
> > While we did not elaborate on this further in the paper, it is possible that the Bayesian approach of [1] can be improved with causal techniques, for example by using the backdoor adjustment (see paragraph “{Leveraging a Causal Model for OPE}”). Sounds like a great research hypothesis to investigate.
> >
> > **“An additional question I have is whether discovering the causal structure of the environment is feasible in the strict POOO regime? If not, then this strictly necessitates pre-specifying the causal graph?”**
> >
> > As we stated after proposition 2, our analysis does not assume that a causal model is available to the learner. It gives conditions under which causal probabilities can be correctly estimated from conditional probabilities regardless of what the true causal model is. We agree it is important to clarify that we do not assume that the true causal model is given or discovered, so we added this to the introduction.
> >
> > Of course if a causal model is given or discovered, it may be leveraged for RL in various ways. We discuss these under related work. Related work also discussed the prospects for learning a causal graph in the POOO regime.
> >
> > **“The paper is extremely long. The core results could be presented in a shorter fashion. Several tables and figures, as well as examples take up much room and could potentially be cut.”**
> >
> > As we mentioned, we have cut 3 pages. We moved the figure illustrating online and on-policy learning to the appendix. We are happy to take advice about examples, but would argue that worked examples are very helpful for readers who want to learn a new concept, and can be skipped by those know the concept already. Our suggestion above of splitting the paper in two may be another way of addressing your concern without sacrificing comprehensiveness.
> >
> > **“the "related work" section takes up 5 and a half pages, which is enormous for a paper that does not aim to be a survey paper.“**
> >
> > We do not aim at a survey paper on causality and RL (a big task taken up by others), but we did aim to conclude the paper with a survey of related work on causal models, and future research directions. Our survey leverages the concepts and insights developed in our main body, especially the importance of the information asymmetry between the behavioral agent and the learning agent. Again we are happy to take advice, but surveys of related work are among the most useful things for readers who want to work in an area.
> >
> > **“Similarly, 7 1/2 pages of background is very thorough for a non-survey paper.”**
> >
> > We have reduced the background on  MDPs by 2.5 pages, mainly by streamlining the presentation.
> >
> > 1.	We directly introduce factored POMDPs, skipping MPDS
> > 2.	We directly introduce factored POMDP polices based on belief states, skipping policies based on observation sequences
> > 3.	We move the derivation of belief updates for factored POMPDs to the appendix, following your suggestion to cut well-known results.
> >
> > POMDP theory is complex and can, in our experience, be a real barrier for non RL researchers, like causality theorists, so we kept the examples. Also our formulation of a factored POMDP where the states are described variables is actually novel (compare with Russell and Norvig’s treatment). So while we develop the theory in parallel with the standard formulation where states emit observation signals, we wanted to keep a complete sequence of the new definitions of factored POMDPs, with explanations of novel aspects.
> >
> > **“I found myself getting somewhat lost while reading it. With a shorter paper that reaches its conclusion quicker, its easier for readers to follow”**
> >
> > This is great feedback thank you. As explained above, we have added summaries at the beginning and end of each section, and an overview figure connecting the sections in the introduction. Also as explained above, we have rewritten the introduction to explain our main conclusion and how they are relevant to RL more quickly.

---

### Review · Reviewer_qKQ2 · 2025-01-05

**Summary Of Contributions:**

Recent folklore suggests that richer approaches to reinforcement learning (RL) invoke models, and *causal* models in particular. But thus far it has not been completely clear *why*, *when*, and *to what extent* causal modeling might be necessary or useful in RL settings. The authors aim to answer this question, and conclude that in RL settings that are partially observable, AND offline, AND off-policy, causal modeling is necessary to estimate value and optimal behavioral policies, since conditional probabilities are not sufficient for estimating them.

The authors also include lengthy tutorial material on relevant causality and RL concepts, including causal Bayesian networks (CBNs), partially observable Markov decision processes (POMDPs), and structural causal models (SCMs).

**Audience:**

Yes

**Broader Impact Concerns:**

No concerns.

**Claims And Evidence:**

Yes

**Requested Changes:**

- Much tutorial material and many parenthetical comments should probably be moved to appendices, as they seriously impede the flow of the paper.
- It would be helpful if tutorial material were more clearly separated from results material about when interventional probabilities equal (or do not equal) conditional probabilities. For example, tutorial material on SCMs follows and precedes some results material.
- Please include an appendix where all of the main results are clearly stated.
- To help the reader make sense of the various Bellman equations that appear, it could be helpful to annotate them better (one- or two-word text descriptions next to each equation, e.g., state value next to V, state-action value next to Q) and to collect all of them in some technical appendix, so it is easier for readers to see their similarities and differences.
- Please address the formatting/typo issues mentioned above.
- Please reduce or combine some goalie diagram instances, and reduce the overall number of main text figures.
- Please do not split the paper into two shorter ones (mentioned in reviewer responses). I think this would only make existing problems worse, and I also do not think both papers would be content-rich enough to stand on their own. The problem is not length due to too much content, but length due to low information density.

**Strengths And Weaknesses:**

This is a frustrating paper. On the one hand, the key question—when and to what extent is causal modeling relevant to RL settings?—is timely and important. On the other hand, the way it is addressed here is meandering, a bit confusing, and appears to be missing a lot of the components I would want out of an answer to that question.

The paper is billed both as a "short" tutorial (pg. 1), and as a statement of new results clarifying the aforementioned question, which I think is overall a weakness. There is too much tutorial material, and it is poorly organized, in the sense that it is easy for a reader not well-acquainted with this material to get lost. The authors say too many things that are not necessary for the main flow of the paper (and could be moved to SI material), which contributes to this issue. The new results are not clearly collected and stated somewhere (e.g., a technical appendix stating novel results), but scattered throughout. The core results themselves are kind of confusingly stated (e.g., Fig. 2 is misleading).

**Strengths**

The authors are clearly experts in the area and competently present a large volume of technical material helpful for understanding the overall problem (when is causal reasoning necessary in RL?), as well as their attempt at a solution to it. Their conclusions appear to be technically sound. The paper will definitely be useful to people trying to understand how (and why) to integrate RL with causal reasoning.

**Major weaknesses**

- There is a big 'forest and trees' issue. For example, there is a lot of intro material on, e.g., CBNs (4 pages), POMDPs (5 pages), and SCMs (6 pages). As the authors go on, I don't think it's clear to readers unacquainted with this material why they're treating each topic, and what each topic contributes to the bigger picture. The goal is to formulate the standard RL POMDP setting in a way that allows causal reasoning to be applied, and in particular in a way that involves some kind of graph structure. Dynamic decision networks (DDNs), which are essentially just the idea of treating the transition model as defining a graph, is the way the authors address this issue, but this fact (core to one's ability to apply causal reasoning to RL!) I find kind of buried in the main-text discussion. I think where DDNs are first introduced (pg. 5), it is worth commenting *why* they are necessary to introduce. Point out earlier what it takes until pg. 10 to mention—that it is not a priori obvious how to port causal insights to RL settings.
- In the authors' recent reviewer responses, they describe their paper as "long but not dense". I disagree, and find it both long and dense. It is not dense in the sense of being technically challenging, but dense in the sense of being kind of confusing and filled with a lot of superfluous or distracting material. There is lots and lots of repetition (in figures, in examples, in equations, ...), probably much of it unnecessary. Contributes to 'forest and trees' problem and makes it hard to understand what the important points are. Would be helpful to move lots of such material to appendices, e.g., "The interested reader can see more on how CBNs are related to SCMs in Appendix XXX." (I claim that this would be *helpful*, not *hurtful*, to the learner. Learners do not know in advance which material is core and which is peripheral, so it is helpful to tell them.)
- Tutorial material is not clearly separated from material discussing new results, which makes it harder to follow both aspects of the paper.
- I'm not convinced that the soccer example (the main one used throughout) is ideal, since it lacks the richness of most RL problems—it's basically a bandit problem, one step and one action. Other examples in the paper (e.g., car braking) have the same issue. The paper is probably too mature to change this, but it would be nice to at least gesture at a richer toy example where the authors' conclusions are relevant.
- The goalie diagram appears, with little modification, a total of ~ 15 times in the main text. I'm fairly confident many of these instances could be consolidated. For example, in several cases (e.g., Figs 13 and 14, Figs 5 and 6) nearly identical figures appear right next to each other and make almost identical points. These could probably be combined (ideally, not by just putting 4 nearly identical graphs in one plot, but by suitably labeling a smaller number of graphs). I think this would help, and not hurt, the naive reader's understanding of the relevant concepts.
- The more I think about it, the more I find the flowchart depicted in Fig. 2 confusing. Whether RL is on/off-policy is independent of whether the state space is fully observable; they are orthogonal axes of variation for RL settings. The flowchart could easily be redone with "observability, yes or no?" as the first decision tree question. To me, this indicates maybe the sequential decision tree isn't the best way to illustrate the core finding. As far as I can tell, what the authors have actually found is that each of on-policy, online, and fully observable is individually sufficient. Why not a schematic that explicitly indicates that each of these features is on equal footing?
- I find the discussion of related work in Sec. 8 a bit meandering. It currently reads like 'here's a bunch of stuff tangentially related to causal RL'. At least to me, given the thrust of the paper, discussing related work related to two questions is most pressing. (1) When, given previous/related work, does causal RL help in practice? (And how is this consistent with the theoretical argument presented here?) (2) How might one actually learn a causal model? Some of the material (e.g. the counterfactual regret section) seems a bit indulgent.

**Minor weaknesses**

- This isn't necessarily a weakness but it did confuse me. In Fig. 7b, can't (WLOG) the latest observation be included in the belief b_t?
- The transition model depicted in Fig. 8 is confusing. I assume (given its in-text description) it is intended to describe a model where multiple shots are made. But there is no arrow that goes from the CG = 0 state to the CG = 1 state.
- pg. 3, Go example. Kind of a nitpick, but there is indeed partial observability in the sense that the opponent's internal variables cannot be observed by the agent. Part of competitive games is modeling one's opponent, since this generally confers a strategic advantage above and beyond being able to observe the board state.


**Formatting/typo issues**

- Fig 18 is referred to 3 times on pg. 3. This could probably be reduced to once or twice.
- Some subsection headings (e.g., 2.2, 3.2) end in periods, while others (e.g., 2.1, 3.1) don't. This doesn't matter much, but uniformity would be nice.
- pg. 13, typo "give several exampes"
- In a few places, not sure if the intent is to put equation references in parentheses. For example, pg. 13, "the behavioral policy Equation (3) of shooting" looks like it ought to have "Equation (3)" in parentheses.
- It's a little bit weird that the two sentences which contain "given a true causal model..." are exactly repeated twice, once in the intro and once just before Sec. 5.2 (pg. 18). Would be better to rephrase or remove one instance.
- pg. 20, "RL settings With..." heading. "Settings" should be capitalized and "with" should not be.
- pg. 20, there should be a comma in the fictitious scenario. "It must be ... than theirs," he thinks.
- Conclusion: The citation to Pearl just says (2000), and should probably also include Pearl's name, since it is nowhere nearby.
- pg. 31, "Sec.7"-> should add a space so it becomes "Sec. 7". There are similar instances elsewhere (e.g., "Thm.1", "Fig.3b").

---

> ### Author Response · Authors · 2025-01-09
> **Replies to General Comments**
>
> We give brief replies to general comments. Line-by-line replies will be given when we finalize our revision.
>
> **Recent folklore suggests that richer approaches to reinforcement learning (RL) invoke models, and causal models in particular.** This is a great way to explain our goal: to formalize and make precise the folklore claims and bring out the assumptions that do or do not justfy them. We have given by now 4 talks with our results and we feel the community appreciates this clarification.
>
> **The more I think about it, the more I find the flowchart depicted in Fig. 2 confusing. Whether RL is on/off-policy is independent of whether the state space is fully observable; they are orthogonal axes of variation for RL settings. The flowchart could easily be redone with "observability, yes or no?" as the first decision tree question. To me, this indicates maybe the sequential decision tree isn't the best way to illustrate the core finding. As far as I can tell, what the authors have actually found is that each of on-policy, online, and fully observable is individually sufficient. Why not a schematic that explicitly indicates that each of these features is on equal footing?**
>
> It's interesting that you find Fig. 2 confusing. Your summary of it is exactly right "what the authors have actually found is that each of on-policy, online, and fully observable is individually sufficient.". We show the disjunction of sufficient conditions in section 5.3 in the quote environment. The advantage of the decision tree is that it shows both the sufficient conditions for causal and conditional policy values to co-incide *and* the case where they do not (the POOO setting), bottom right leaf. In our talks we have used a simple table, this is probably what you mean by "linear schematic", we will try that.
>
> **I find the discussion of related work in Sec. 8 a bit meandering. It currently reads like 'here's a bunch of stuff tangentially related to causal RL'. At least to me, given the thrust of the paper, discussing related work related to two questions is most pressing. (1) When, given previous/related work, does causal RL help in practice? (And how is this consistent with the theoretical argument presented here?) (2) How might one actually learn a causal model? Some of the material (e.g. the counterfactual regret section) seems a bit indulgent.** This is a helpful comment, thank you. We thought we were summarizing previous work on when causal RL helps in practice but we don't seem to have made this clear. P. 29 gives the three advantages of causal models over non-causal models (resolving confounding, support counterfactuals, modelling local relationshiops),  and listed work that uses these. We will try to tie the discussion more clearly to these advantages. Also we will remove suggestions for future research---perhaps this is what you mean by "feels indulgent"---and that will also make the section shorter. As for counterfactual regret, we are inclined to agree with you, but Barenboim has emphasized this point in his tutorial and several papers, including recent ones in prominent venues (e.g., neurips). We will shorten our discussion of this work to a mention.
>
> As we said, line-by-line replies to your general comments will come in the revision. Thank you again for the thought you have put into our submission.

---

> ### Author Response · Authors · 2025-01-22
> **Line-by-line Replies to go with our new revision 1/2**
>
> Thank you for the great and detailed suggestions! The paper organization suggestions were especially helpful.
>
> As before new material added in response to reviewer comments appears in red.
>
> **I think where DDNs are first introduced (pg. 5), it is worth commenting why they are necessary to introduce. Point out earlier what it takes until pg. 10 to mention—that it is not a priori obvious how to port causal insights to RL settings.**
>
> We now explain when DDNs are introduced in Section 1 (p.5) why we need a causal model that can represent POMDPs. At the beginning of Section 3 where DDNs are formally defined, we again explain why they are necessary for defining causal concepts relevant to RL.
>
> **Would be helpful to move lots of such material to appendices, e.g., "The interested reader can see more on how CBNs are related to SCMs in Appendix XXX."**
>
> We have implemented your suggestion of using a "main text+appendix" structure, which we believe has improved the presentation a lot. The main body of the text minus appendices is now down to just over 19 pages.
>
> **Tutorial material is not clearly separated from material discussing new results, which makes it harder to follow both aspects of the paper.**
>
> Tutorial material is now in appendix.
>
> **I'm not convinced that the soccer example (the main one used throughout) is ideal, since it lacks the richness of most RL problems—it's basically a bandit problem, one step and one action.**
>
> We agree that the causal RL literature has focused on bandit problems especially in examples. Table 7 and appendix F give examples of policy evaluation in a sequential setting. Briefly, the dynamic sports example of Figure 4 models a situation where if a team does not take a shot, with some probability they maintain position and can attack again. We solve for the value functions in this sequential scenario.
>
> **The goalie diagram appears, with little modification, a total of ~ 15 times in the main text.**
>
> The goalie diagram now appears only once in the main text. In the appendix, there are three more goalie figures to illustrate computing causal reward probabilities. They are consolidated into a single appendix section.
>
> **As far as I can tell, what the authors have actually found is that each of on-policy, online, and fully observable is individually sufficient. Why not a schematic that explicitly indicates that each of these features is on equal footing?**
>
> We have replaced the decision tree by a table where each row corresponds to an RL setting.
>
> **At least to me, given the thrust of the paper, discussing related work related to two questions is most pressing. (1) When, given previous/related work, does causal RL help in practice? (And how is this consistent with the theoretical argument presented here?) (2) How might one actually learn a causal model? Some of the material (e.g. the counterfactual regret section) seems a bit indulgent.**
>
> We have removed the counterfactual regret section. And focused on the uses of causal RL in different RL tasks. We also kept the sections on learning causal models. Using your guidance, we have cut the related work section from over 5 pages to under 4 pages.
>
> **In Fig. 7b, can't (WLOG) the latest observation be included in the belief b_t?**
>
> If you mean that $b_t$ includes the observation $o_t$, then yes in the sense that $b_t$ is a posterior over latent variables conditional on $o_t$. POMDP notation is very compact in that way. We have written out the dependence explicitly in Appendix B.2. In general it is important to separate observations from beliefs so that you can model how observed variables like player health cause actions like shooting. We comment on this after Definition 1 of DDNs.
>
> **The transition model depicted in Fig. 8 is confusing. I assume it is intended to describe a model where multiple shots are made. But there is no arrow that goes from the CG = 0 state to the CG = 1 state.**
>
> As explained in the context of the DDN (now section 3.2), CG=0 is an absorbing state: once the team loses possession, the attack is over. We have added this comment to the caption of Figure 11 (formerly Figure 8). Also the figure is now in the appendix.
>
> **Go example. Kind of a nitpick, but there is indeed partial observability in the sense that the opponent's internal variables cannot be observed by the agent.**
>
> This came up with the audience at one our talk. We hope the general point is clear, that if one does treat an RL problem as one of perfect observability, there is no issue with latent variables. One could model a perfect information boardgame as partially observable as you write. However, we mention AlphaZero which does not do opponent modelling, and the same goes for other state of the art programs like Stockfish for chess. What is nice about the AlphaZero example is that it featured an early phase of offline learning, but confounding issues did not need to be addressed because they treated the game like a completely observable one.

---

> ### Author Response · Authors · 2025-01-22
> **Line-by-line Replies to go with our new revision 2/2**
>
> **Fig 18 is referred to 3 times on pg. 3. This could probably be reduced to once or twice.**
>
> Indeed. Now only cited once (as Figure 2).
>
> **Some subsection headings (e.g., 2.2, 3.2) end in periods, while others (e.g., 2.1, 3.1) don't. This doesn't matter much, but uniformity would be nice.**
>
> All periods are removed.
>
> **pg. 13, typo "give several exampes"** fixed
>
> **In a few places, not sure if the intent is to put equation references in parentheses. For example, pg. 13, "the behavioral policy Equation (3) of shooting" looks like it ought to have "Equation (3)" in parentheses.**
>
> We rewrote the particular passage. We added parentheses to several places where equations were cited in this way.
>
> **It's a little bit weird that the two sentences which contain "given a true causal model..." are exactly repeated twice, once in the intro and once just before Sec. 5.2 (pg. 18). Would be better to rephrase or remove one instance.**
>
> We rephrased because another reviewer said it was important to clarify in the introduction whether we were assuming that a causal model is given.
>
> **pg. 20, "RL settings With..." heading. "Settings" should be capitalized and "with" should not be.**  thanks fixed
>
> **pg. 20, there should be a comma in the fictitious scenario. "It must be ... than theirs," he thinks.**
> thanks fixed
>
> **Conclusion: The citation to Pearl just says (2000), and should probably also include Pearl's name, since it is nowhere nearby.**
>
> We just dropped the citation
>
> **pg. 31, "Sec.7"-> should add a space so it becomes "Sec. 7". There are similar instances elsewhere (e.g., "Thm.1", "Fig.3b").**  we dropped the Sec. 7 reference and checked for the abbreviations elsewhere.
>
> **Requested Changes:**
>
> **Much tutorial material and many parenthetical comments should probably be moved to appendices, as they seriously impede the flow of the paper.**
>
> Great suggestion, we implemented it, see above and the general comments to go with the revision.
>
> **It would be helpful if tutorial material were more clearly separated from results material about when interventional probabilities equal (or do not equal) conditional probabilities. For example, tutorial material on SCMs follows and precedes some results material.**
>
> Tutorial material is now in appendix as you suggested above.
>
> **Please include an appendix where all of the main results are clearly stated.**
>
> After streamlining the paper (see revision comments), there is now only one main result (Theorem 1). It is stated also in the appendix D.1 with a proof.
>
> **To help the reader make sense of the various Bellman equations that appear, it could be helpful to annotate them better (one- or two-word text descriptions next to each equation, e.g., state value next to V, state-action value next to Q) and to collect all of them in some technical appendix, so it is easier for readers to see their similarities and differences.**
>
> We added annotations to the what-if Bellman equation in the text (Section 4.1). Appendix C collects the observational and interventional Bellman equations as you suggest. The observational Bellman equation is the standard textbook one for belief MDPs. We show how the interventional and what-if Bellman equations are derived from the observation Bellman equation.
>
> **Please address the formatting/typo issues mentioned above.**
>
> done
>
> **Please reduce or combine some goalie diagram instances, and reduce the overall number of main text figures.**
>
> See above, only 1 goalie diagram and 5 figures overall remain in the main text.
>
> **Please do not split the paper into two shorter ones (mentioned in reviewer responses). I think this would only make existing problems worse, and I also do not think both papers would be content-rich enough to stand on their own. The problem is not length due to too much content, but length due to low information density.**
>
> Thank you for the advice which we took.

---

### Author Response · Authors · 2024-12-31
**Broader Impact Statement**

The paper is a theoretical study of the causal foundations of reinforcement learning settings. We do not see a direct broader impact beyond  AI foundations. We would be happy to add a statement to that effect.

---

### Author Response · Authors · 2025-01-09
**Response to suggestions. 1: requested changes**

Thank you for the suggestions. We appreciate that you have given a lot of thought for how to improve our paper structure. The action editor  asked us to "respond soon" to support discussion. We are therefore posting a quick reply with our reactions and plans to respond. As you have made requests for substantive changes----which is good---it will take us a while to impement the changes, probably a week, but we wil try to do it asap.

Let's start with the requested changes going line-by-line.

**Requested Changes:
Much tutorial material and many parenthetical comments should probably be moved to appendices, as they seriously impede the flow of the paper.
It would be helpful if tutorial material were more clearly separated from results material about when interventional probabilities equal (or do not equal) conditional probabilities. For example, tutorial material on SCMs follows and precedes some results material.**

A more extensive use of appendices sounds like a good solution. We do need to give the definitions required by our theorems. We're thinking what you refer to as tutorial material are examples and comments to explain the definitions. Our plan is to move those to the appendix, or at least any explanation/example that takes more than a couple of lines.

**Please include an appendix where all of the main results are clearly stated.** Sure no problem.

**To help the reader make sense of the various Bellman equations that appear, it could be helpful to annotate them better (one- or two-word text descriptions next to each equation, e.g., state value next to V, state-action value next to Q) and to collect all of them in some technical appendix, so it is easier for readers to see their similarities and differences.**

That's fine. The main thing we are thinking of though is to have a *single* Bellman equation based on what-if counterfactuals. As we explained on p.23 (below equation 18), interventional and observational probabilities are special cases of what-if counterfactuals (a point Pearl makes often). We can spell out the special cases in the appendix. A single Bellman equation would reduce the main body length and increase the information density as you are recomending we do.

**Please address the formatting/typo issues mentioned above.** No problem thank you.

**Please reduce or combine some goalie diagram instances, and reduce the overall number of main text figures.** We will try to use Figure 10 only, and move the other examples to appendices or remove them altogether. Figure 10 is the only dynamic multi-step example. Focusing on it would address your other comment: "I'm not convinced that the soccer example (the main one used throughout) is ideal, since it lacks the richness of most RL problems—it's basically a bandit problem, one step and one action."

**Please do not split the paper into two shorter ones (mentioned in reviewer responses). I think this would only make existing problems worse, and I also do not think both papers would be content-rich enough to stand on their own. The problem is not length due to too much content, but length due to low information density.** Thank you for giving a clear answer to our suggestion. We are thinking of an intermediate approach: state the results in terms of what-if counterfactuals and note that they cover interventional probabilities as a special case (with example illustrations in the appendix). This would go with starting with structural causal models and moving causal Bayesian networks in the appendix. This is similar to the approach taken by Scholkopf and Bengio where SCMs are the main definition and other causal models are viewed as special cases. Leaving the relationship between Bayesian networks and SCMs in the appendix is also an example of "moving tutorial material to the appendix" as you suggest.

---

> ### Comment · Reviewer_UboP · 2025-01-09
> **Comment**
>
> Dear authors,
>
> Thanks for the updates! Reviewing the updated draft, I noticed one thing that I overlooked previously. However, I think it contributes massively to the feeling that several reviewers mentioned that the paper reads as "busy" or is structured strangely. Many figures in your draft are inlined instead of floating to the top or the bottom of the page. I don't know if the TMLR template specifies this, but it is non-standard for most venues that I know of. The main problem is that the figures interrupt the reading flow in an awkward manner.
>
> Can you upload an update where you using floating mechanisms for the figures? I genuinely believe this will improve the overall readability quite drastically.
>
> You mentioned the final typesetter, but I don't think TMLR has another typesetting round like other journals do? Maybe the AC can comment.

---

> > ### Author Response · Authors · 2025-01-10
> > **Figure placement**
> >
> > Hi sure, we can make figures float to the top or bottom. Reviewer qKQ2 recommended removing figures and/or moving them to the appendix in any case so fewer figures to worry about.

---

> ### Comment · Reviewer_qKQ2 · 2025-01-09
>
> There is no final typesetter other than the authors. The version the authors have posted when the paper is accepted is the final version.

---

> > ### Author Response · Authors · 2025-01-10
> > **final typesetting**
> >
> > Okay thanks for clarifying that final versions are published "as is" after acceptance. Different from other journals in our experience but sure no problem.

---

> > > ### Author Response · Authors · 2025-01-22
> > > **finalized figure placement**
> > >
> > > Our latest revision has all the figures set with the [tb] option as the reviewer has suggested. Also we checked that no figures float from one section to the other.

---

### Author Response · Authors · 2025-01-20
**Hold on before submitting acceptance/rejection recommendations**

Just a quick message to the reviewers.  Please hold on before submitting your recommendations for acceptance/rejection.  We are finalizing the revisions to the paper.  We will upload the revisions asap and then post another message indicating that you can do your final assessment.

---

### Author Response · Authors · 2025-01-24
**revisions completed**

Just a quick message to indicate that we completed our revisions, uploaded the revised draft with material changes highlighted in red and posted line by line responses to each review.  We thank the reviewers for all the feedback.  Feel free to proceed with your evaluation as well as recommendations for acceptance/rejection.  If there is anything else you'd like us to address, do not hesitate to post a message and we will be happy to do further revisions.

---

> ### Author Response · Authors · 2025-03-15
> **Is there an update on a decision**
>
> Hello TMLR, we completed our last revision on January 23, 2025 (see comment). A couple of questions:
>
> * are there any action items for us now? We believe not but please let us know if we missed something.
> * is there a timeline for when we should expect a decision on acceptance?
>
> thank you, the authors

---

### Decision · Action_Editor_4L2e · 2025-03-28

**Recommendation:** Accept with minor revision

**Comment:**

We strongly suggest that the authors revise the paper based on recommendations by reviewers and try to make sure they're all accounted for. In particular, reviewer UboP still believes there is room for growth in presentation for this work to be most accessible to a new audience.

**Audience:**

Given that this is a tutorial style paper, I can see people in TMLR audience being interested in this paper, especially since there is a growing interest in the intersection of causality and RL.

**Claims And Evidence:**

The claims made in the submission are accurate, convincing, and clear in their evidence.

---

> ### Author Response · Authors · 2025-04-16
> **acknowledgement of decision**
>
> Dear Action Editor:
>
> thank you for the positive decision! We are working on the camera-ready version and are also planning to post a short video as suggested by the editors.

---

> ### Author Response · Authors · 2025-04-24
> **cannot see new recommendations by the reviewers**
>
> Dear action editor, we are trying to revise the paper based on your request.  However, we cannot see the new recommendations by the reviewers.  All we can see are the reviews that they submitted back in January.  You wrote "reviewer UboP still believes there is room for growth in presentation for this work to be most accessible to a new audience", but the last edit by reviewer UboP dates from January 6.  Similarly, you wrote "We strongly suggest that the authors revise the paper based on recommendations by reviewers and try to make sure they're all accounted for", however all we can see are the reviews that were last modified in January and we already addressed those recommendations.   We believe the system does not allow us to see any recommendation that was posted after the rebuttal period.  Those recommendations are probably only viewable by the action editor.  Hence, can you copy-paste those recommendations or summarize them so that we can act on them.